# Instruction Lens Score: Your Instruction Contributes a Powerful Object Hallucination Detector for Multimodal Large Language Models

**Runhe Lai** [1 2 3 *]  **Xinhua Lu** [1 2 3 *]  **Yanqi Wu** [1 2 3]  **Jinlun Ye** [1 2 3]  **Weijiang Yu** [1 †]  **Ruixuan Wang** [1 2 3 †]

## Abstract

Multimodal large language models (MLLMs) have achieved remarkable progress, yet the object hallucination remains a critical challenge for reliable deployment. In this paper, we present an in-depth analysis of instruction token embeddings and reveal that they implicitly encode visual information while effectively filtering erroneous information introduced by misleading visual embeddings. Building on this insight, we propose the Instruction Lens Score (InsLen), which combines a Calibrated Local Score with a Context Consistency Score that measures context consistency of the object tokens. The proposed approach serves as a plug-and-play object hallucination detector without relying on auxiliary models or additional training. Extensive experiments across multiple benchmarks and diverse MLLM architectures demonstrate that InsLen consistently outperforms existing hallucination detection methods, highlighting its effectiveness and robustness. The code is available at https://github.com/Fraserlairh/Instruction-Lens-Score.

## 1. Introduction

In recent years, Multimodal Large Language Models (MLLMs) have demonstrated impressive abilities in multimodal understanding, reasoning, and interaction, enabling a wide range of real-world applications. However, the object hallucination (OH) issue in MLLMs significantly undermines their reliability in real-world deployment, where

---

*Equal contribution [1]School of Computer Science and Engineering, Sun Yat-sen University, Guangzhou, China [2]Peng Cheng Laboratory, Shenzhen, China [3]Key Laboratory of Machine Intelligence and Advanced Computing, MOE, Guangzhou, China. Correspondence to: Weijiang Yu <yuwj39@mail.sysu.edu.cn>, Ruixuan Wang <wangruix5@mail.sysu.edu.cn>.

*Proceedings of the 43rd International Conference on Machine Learning*, Seoul, South Korea. PMLR 306, 2026. Copyright 2026 by the author(s).

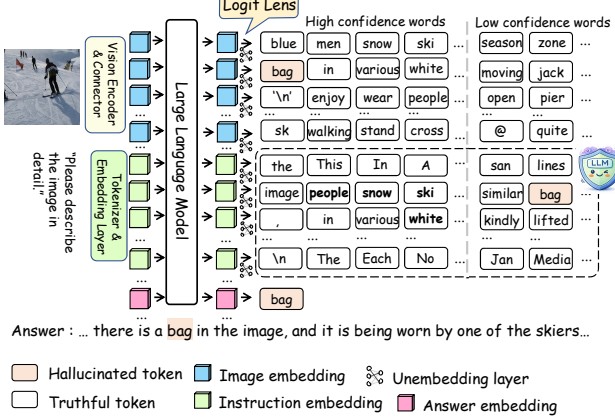

*Figure 1.* Illustration of instruction embeddings filtering misleading visual information. By applying the Logit Lens to intermediate embeddings, we observe that instruction embeddings (green) consistently assign higher confidence to image-grounded concepts (e.g., people, snow, and ski), while suppressing hallucinated objects (e.g., bag). See more qualitative examples in Appendix A.4.

models generate responses that are inconsistent with the actual objects present in the input images (Gunjal et al., 2024). Therefore, detecting hallucinated objects in the responses generated by MLLMs has become a crucial component for building reliable AI systems (Zhu et al., 2025).

Some studies (Liu et al., 2024a; Jing et al., 2024; Park et al., 2025a) detect OH with auxiliary models such as GPT-4 (Achiam et al., 2023). However, their deployment comes with considerable computational overhead. Another line of research (Zhou et al., 2024; Jiang et al., 2025a;b; Park & Li, 2025; Che et al., 2025) analyzes potential causes of OH by examining intermediate outputs of the model. In particular, visual information has become a central cue for detection, including attention weights over visual tokens (Jiang et al., 2025b) and similarity between textual answer embeddings and patch-level image embeddings (Park & Li, 2025). However, these vision-based scores can be adversely affected by misleading visual information introduced by the visual encoder (Gong et al., 2024; Luo et al., 2025) or during cross-modal interactions within the language model (Seo et al., 2025), which may result in spuriously high confidence for

hallucinated objects. To address these limitations, we delve into the embeddings of instruction tokens and reveal a key finding, i.e., *misleading information introduced by visual embeddings can be effectively filtered out by instruction embeddings.* As shown in Figure 1, the Logit Lens technique (nostalgebraist, 2020) is employed to interpret the embeddings corresponding to image patches and instruction tokens in MLLM's vocabulary. We observe that part of the instruction embeddings encodes object-related visual cues grounded in the image alongside the prediction of instruction tokens. Meanwhile, hallucination arising from misleading visual embeddings (e.g., the "bag" in Figure 1) is often assigned low confidence in the instruction embeddings. In Section 4.1, a statistical analysis is also provided to support our observation. These findings highlight an overlooked capability of instruction embeddings for detecting object hallucination.

Building on this insight, we design an InsLen score for object hallucination detection, where instruction embeddings are used both to calibrate misleading visual information and to provide object-related contextual cues. Accordingly, the InsLen score consists of two complementary components: (1) Calibrated Local Score, which calibrates spurious high-confidence predictions from vision-based scores; (2) Context Consistency Score, which measures the consistency between the embeddings of generated objects in the textual answer and object-related instruction embeddings.

Specifically, the Calibrated Local Score introduces a Calibration Confidence by measuring the maximum confidence assigned to the object answer token across instruction embeddings, thereby leveraging the filtering effect of instruction embeddings. This confidence is combined with vision-based scores adopted in previous work (Jiang et al., 2025a;b; Park & Li, 2025) to calibrate local visual evidence. On the other hand, existing vision-based scores are derived from patch-level image embeddings, where local visual patterns may overlap between real and hallucinated objects (e.g., similar silver materials for spoons and knives), making it difficult to distinguish hallucinated objects that visually resemble real ones. To address this limitation, we leverage instruction embeddings, which are computed conditioned on all image embeddings through cross-modal attention, and capture the global information about objects. Therefore, instruction embeddings that assign high confidence to the object token are collected to provide global context cues about the object token. The Context Consistency Score then evaluates whether the generated object in the answer output is consistent with such global object context encoded in instruction embeddings, which helps distinguish hallucinations caused by similar local visual patterns from true objects. Combining the two scores, our InsLen score jointly captures calibrated patch-level visual evidence and global contextual consistency, enabling more reliable detection of

hallucinated objects. The main contributions of this paper are summarized as follows.

- An analysis of instruction embeddings in MLLMs is conducted and reveals a key finding that misleading information introduced by visual embeddings can be effectively filtered out by instruction embeddings.

- The Instruction Lens Score is proposed, which integrates a Calibrated Local Score with a Context Consistency Score for object hallucination detection.

- Extensive experiments are conducted from multiple perspectives, demonstrating the effectiveness and robustness of the proposed InsLen score on object hallucination detection benchmarks.

## 2. Related Work

**Multimodal Large Language Models.** Building on the rapid progress of Large Language Models (LLMs), such as LLaMA (Touvron et al., 2023) and Mixtral (Jiang et al., 2024), recent studies have increasingly extended language-centric reasoning to multimodal settings, resulting in various multimodal large language models (MLLMs). LLaVA-1.5 (Liu et al., 2024b) and InstructBLIP (Dai et al., 2023) are widely used open-source representatives of MLLMs, while more recent MLLMs have incorporated advanced techniques (e.g., DeepStack (Meng et al., 2024)), including mPLUG-Owl3 (Ye et al., 2025), Qwen3-VL (Bai et al., 2025), and LLaVA-OneVision1.5-Instruct (An et al., 2025). However, hallucination remains a persistent challenge for MLLMs, which can substantially impair their reliability across a variety of vision–language tasks (Bai et al., 2024).

**Object Hallucination Detection and Mitigation.** Object hallucination issue (OH) refers to the phenomenon that MLLMs generate descriptions of objects that do not exist in the visual input. This issue poses significant risks to the reliability of MLLM-based systems, especially in safety-critical applications such as medical imaging (Zhu et al., 2025). One line of approaches aims to mitigate hallucination through improved training paradigms (Yu et al., 2024; 2025) and decoding-time interventions (Deng et al., 2024; Leng et al., 2024; Park et al., 2025b; Suo et al., 2025; Chen et al., 2024). In particular, contrastive decoding methods (Leng et al., 2024; Park et al., 2025b; Chen et al., 2024) reduce hallucinations by contrasting outputs generated under different visual conditions or decoding trajectories, thereby encouraging stronger reliance on visual evidence and suppressing language priors. Another line of approaches propose to detect hallucination, which can be broadly categorized into methods that rely on auxiliary models and those that exploit internal signals of MLLMs. For example, GAVIE (Liu et al., 2024a) leverages GPT-4 (Achiam et al., 2023) to assign

accuracy scores to generated responses, while HalLoc (Park et al., 2025a) constructs a dataset containing hallucinated answers and trains a dedicated hallucination detector. However, these methods incur substantial computational overhead due to additional training procedures and dependence on external models. In contrast, several recent methods detect hallucinations directly from the intermediate output of MLLMs without auxiliary resources (Zhou et al., 2024; Che et al., 2025; Jiang et al., 2025a;b; Park & Li, 2025). LURE (Zhou et al., 2024) estimates token-level uncertainty using the negative log-likelihood of the predicted distribution. Internal Confidence (Jiang et al., 2025a) computes the maximum probability of the object across the image embeddings to confirm whether the object exists in the input image. SVAR (Jiang et al., 2025b) utilizes the attention ratio of the generated token assigned to the image embeddings to measure the contribution of visual information for the generated token. Recently, EAZY (Che et al., 2025) proposes to "zero out" hallucinatory image tokens and identifies hallucinated objects by comparing outputs before and after the zeroing operation. GLSIM (Park & Li, 2025) integrates patch-level visual signals with summarized representations of both the image and the prompt to design a holistic OH detector. Notably, recent approaches benefit substantially from using visual information for OH detection. In this work, we focus on the previously overlooked instruction embeddings and leverage them to further enhance OH detection.

## 3. Preliminaries

**Object Hallucination Detection.** Given an image $I$ and an instruction $\mathbf{X}$ as input, an MLLM generates an output sequence $Y = \{\mathbf{y}_1, ..., \mathbf{y}_T\}$, where $T$ denotes the sequence length. The goal of object hallucination detection is to determine whether each object token $\mathbf{o} \in Y$ corresponds to a real object present in the image $I$ or is hallucinated. To this end, a scoring function $S(\mathbf{o})$ computed from the model outputs is used to assess the reliability of each object token. Formally, the detection result is defined as

$$\mathrm{D}(\mathbf{o}) = \begin{cases} \text{Hallucination,} & \text{if } S(\mathbf{o}) \le \mu, \\ \text{Truth,} & \text{if } S(\mathbf{o}) > \mu, \end{cases} \quad (1)$$

where $\mathrm{D}(\mathbf{o})$ denotes the decision for object token $\mathbf{o}$, and $\mu$ is a predefined threshold used to convert the continuous score $S(\mathbf{o})$ into a discrete classification result.

**Logit Lens.** The Logit Lens technique (nostalgebraist, 2020) aims to interpret intermediate-layer representations in language models by projecting them into the language model's vocabulary. Specifically, Logit Lens. leverages the unembedding matrix $\mathbf{W}_u \in \mathbb{R}^{|\mathcal{V}| \times d}$ of MLLM's language model to map the embedding $\mathbf{h}_l \in \mathbb{R}^d$ from the $l$-th transformer layer into a probability distribution over the vocabulary as $\mathrm{softmax}(\mathbf{W}_u \cdot \mathbf{h}_l) \in \mathbb{R}^{|\mathcal{V}|}$, where $|\mathcal{V}|$ denotes

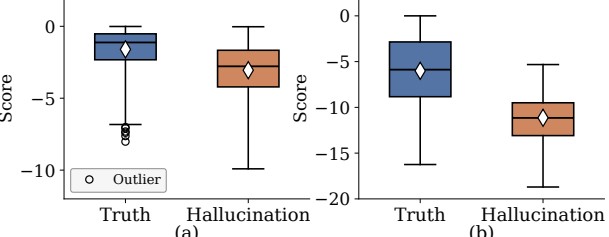

*Figure 2.* Distributions of the log-transformed internal confidence score assigned to hallucinated objects and real objects with LLaVA-1.5, where the MSCOCO dataset is used. (a) Confidence score distributions derived from image embeddings; (b) Confidence score distributions derived from instruction embeddings. More results on different models are provided in Appendix Figure 9.

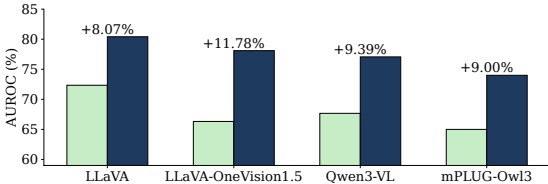

*Figure 3.* Performance comparison of internal confidence derived from image embeddings (green) and instruction embeddings (blue) across different MLLMs on the MSCOCO benchmark.

the vocabulary size of the language model. Tokens with the highest probability are then interpreted as semantic concepts encoded in the corresponding embeddings.

**Local Similarity Score.** The Local Similarity Score (Park & Li, 2025) is proposed to detect object hallucination by evaluating the cosine similarities between the token embedding $\mathbf{h}_\mathbf{o}$ corresponding to the object token $\mathbf{o}$ and the image embeddings associated with the object region, where these image embeddings and object token embeddings are extracted from the $l$-th and $l'$-th decoder layer of MLLM's language model, respectively. Specifically, the image embeddings $\{\mathbf{v}_k\}_{k=1}^K$ of the object region are obtained by selecting the top-$K$ embeddings that assign the highest prediction confidence to the object token $\mathbf{o}$ when the Logit Lens technique is applied to all image embeddings, defined as

$$\{\mathbf{v}_k\}_{k=1}^K = \underset{\mathbf{v}_i \in \mathcal{I}}{\mathrm{Top}\, K} \ \{\mathrm{softmax}(\mathbf{W}_u \cdot \mathbf{v}_i)[\mathbf{o}]\}, \quad (2)$$

where $\mathcal{I}$ denotes the set of image embeddings extracted from the language model, and $[\mathbf{o}]$ denotes the element indexed by the token $\mathbf{o}$. The Local Similarity score $S_{\mathrm{local}}$ is computed as the mean cosine similarity between $\mathbf{h}_\mathbf{o}$ and the selected image embeddings $\{\mathbf{v}_k\}_{k=1}^K$, i.e.,

$$S_{\mathrm{local}} = \frac{1}{K} \sum_{k=1}^K \cos(\mathbf{h}_\mathbf{o}, \mathbf{v}_k). \quad (3)$$

# 4. Method

## 4.1. Motivation

The embeddings corresponding to instructions in MLLMs play an important role in integrating visual information from image embeddings and guiding answer generation. A key finding is that these instruction embeddings effectively filter out misleading information originating from the image embeddings. Specifically, we use the Logit Lens introduced in Section 3 to project both the instruction embeddings and image embeddings into probability distributions over vocabulary. We then compare the confidence assigned to object tokens (which are part of the generated answer output) by the distributions derived from instruction embeddings and image embeddings, respectively. As shown in Figure 2, the confidence score distributions of hallucinated and real objects exhibit noticeable overlap when using image embeddings, whereas confidence derived from instruction embeddings leads to a clearer separation between the two distributions. To further quantify this difference, we evaluate the discriminative performance of the confidence score in distinguishing real objects from hallucinated ones. As shown in Figure 3, confidence computed from instruction embeddings consistently achieves higher performance than confidence derived from image embeddings (by $\geq 8\%$ AUROC). These qualitative results further validate the ability of instruction embeddings to suppress misleading visual information. Therefore, we leverage this property of instruction embeddings to calibrate the local visual evidence (Section 4.2) and to provide oject-related contextual cues (Section 4.3).

## 4.2. Calibrated Local Score

Since vision-based scores can be spuriously overconfident under misleading visual cues, we propose to leverage the visual content encoded in instruction embeddings for additional calibration. To achieve this, a **Ca**librated **Confidence** (Cafe) is introduced to regulate these vision-based scores and enhance the separation between real and hallucinated object tokens, as shown in Figure 4 (Top right). First, for the $j$-th token in the instruction, its embedding $\mathbf{z}_j$ is projected to obtain the probability distribution over the vocabulary using the Logit Lens. Then, for the object token $\mathbf{o}$ in the generated answer, the confidence of the object appearing in the input image can be estimated by the maximum confidence assigned to $\mathbf{o}$ over all instruction embeddings, named the Calibration Confidence score $S_{\text{cafe}}(\mathbf{o})$,

$$S_{\text{cafe}}(\mathbf{o}) = \max_{j=1,...,M} \text{softmax}\{\mathbf{W}_u \cdot \mathbf{z}_j / \tau\}[\mathbf{o}]. \quad (4)$$

Here $\tau$ is a temperature hyperparameter for smoothing the distribution and mitigating overconfident predictions, and $M$ denotes the number of instruction embeddings. Through the filtering effect observed in Section 4.1, hallucinated

object tokens are expected to receive lower calibration confidence. We use this confidence as a calibration factor to fuse with the vision-based score, thereby suppressing spuriously high confidence caused by misleading visual signals. For example, integrating $S_{\text{cafe}}$ with the Local Similarity Score defined in Eq. (3) yields the Calibrated Local Score as

$$S_{\text{cls}}(\mathbf{o}) = S_{\text{cafe}}(\mathbf{o}) \cdot \frac{1}{K} \sum_{k=1}^{K} \cos(\mathbf{h}_\mathbf{o}, \mathbf{v}_k). \quad (5)$$

The multiplicative fusion, rather than additive aggregation, is adopted to ensure compatibility with various vision-based scores such as SVAR score (Jiang et al., 2025b) and Internal Confidence score (Jiang et al., 2025a) without using the extra scaling coefficient.

## 4.3. Context Consistency Score

Vision-based scores extract information from patch-level image embeddings, which are insufficient to provide a global view of the object for detection. By contrast, instruction embeddings are computed conditioned on the image embeddings and the input instruction, allowing attention mechanisms to integrate visual evidence across patches into object-related contextual cues. With this in mind, we propose to identify the instruction embeddings associated with the generated object token $\mathbf{o}$ and evaluate their consistency degree with the embedding of the object token. Specifically, as shown in Figure 4 (Bottom right), each of the instruction embeddings is projected using the unembedding matrix $\mathbf{W}_u$ to obtain the probability distribution over the vocabulary, and the top-$m$ instruction embeddings $\{\hat{\mathbf{z}}_n\}_{n=1}^m$ that assign relatively higher probabilities to the object token $\mathbf{o}$ are selected. These embeddings are then averaged to obtain $\bar{\mathbf{z}} = \frac{1}{m} \sum_{n=1}^m \hat{\mathbf{z}}_n$, which serves as object-related contextual cues. Motivated by the observations in Section 4.1 that instruction embeddings encode grounded object information while suppressing misleading visual information, the embedding $\mathbf{h}_\mathbf{o}$ corresponding to a truthful object token $\mathbf{o}$ is expected to exhibit high context consistency with this averaged embedding. Here, the consistency score between $\mathbf{h}_\mathbf{o}$ and $\bar{\mathbf{z}}$ is defined as

$$S_{\text{con}}(\mathbf{o}) = \alpha - \frac{||\mathbf{h}_\mathbf{o} - \bar{\mathbf{z}}||}{||\mathbf{h}_\mathbf{o}||}, \quad (6)$$

where $|| \cdot ||$ denotes the $\ell_2$ norm. The denominator $||\mathbf{h}_\mathbf{o}||$ is used to normalize the score, and the hyperparameter $\alpha$ ensures that the score remains positive. Using the $\ell_2$ distance allows the metric to capture not only directional differences but also magnitude discrepancies between embeddings. The consistency score $S_{\text{con}}$ is further weighted by the average confidence $\bar{p}(\mathbf{o}|\{\hat{\mathbf{z}}_1,...,\hat{\mathbf{z}}_m\})$ over the selected instruction embeddings, yielding the Context Consistency Score $S_{\text{ccs}}(\mathbf{o})$,

$$S_{\text{ccs}}(\mathbf{o}) = S_{\text{con}}(\mathbf{o}) \cdot \bar{p}(\mathbf{o}|\{\hat{\mathbf{z}}_1,...,\hat{\mathbf{z}}_m\}), \quad (7)$$

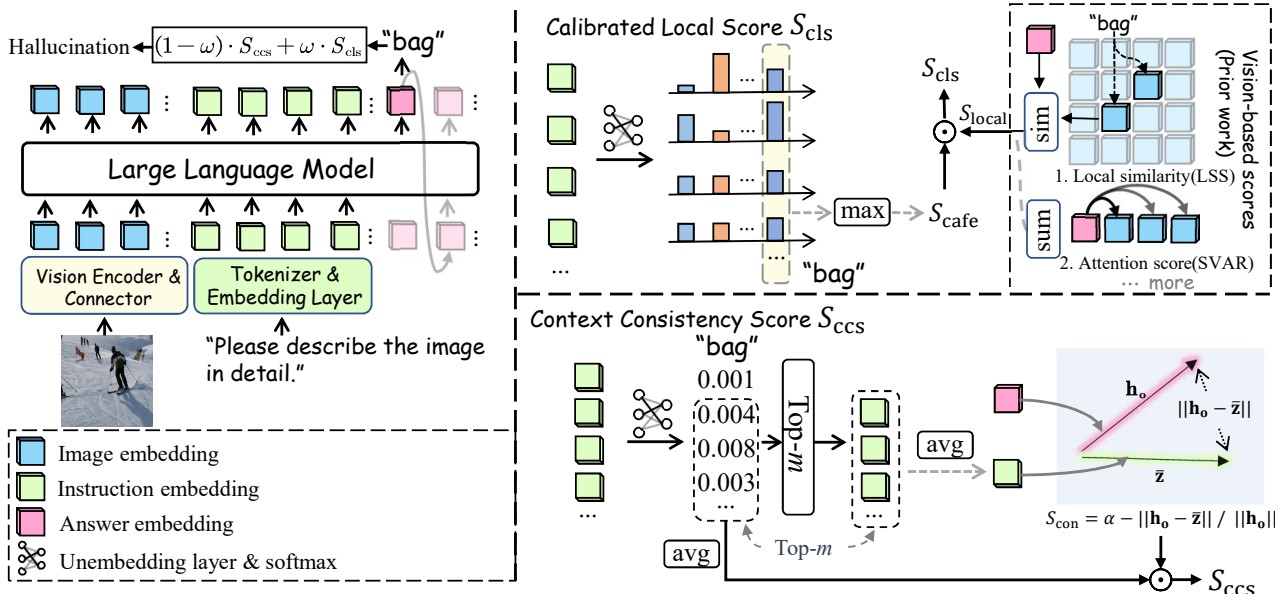

*Figure 4.* Overview of the proposed Instruction Lens score (InsLen) for object hallucination detection. The InsLen score consists of two components: Calibrated Local Score and Context Consistency Score (top left). **Top right** (Calibrated Local Score $S_{cls}$): the maximum confidence $S_{cafe}$ assigned to the object token from instruction embeddings is computed to calibrate the vision-based score $S_{local}$. **Bottom right** (Context Consistency Score $S_{ccs}$): instruction embeddings that assign relatively higher confidence to the object token are aggregated as $\bar{z}$, and their consistency with the generated object embedding is obtained as the Consistency Score $S_{con}$. Finally, Context Consistency Score $S_{ccs}$ weights Consistency Score $S_{con}$ by the average confidence corresponding to the selected instruction embeddings.

where $\bar{p}(\mathbf{o}|\hat{\mathbf{z}}_1, ..., \hat{\mathbf{z}}_m) = \frac{1}{m}\sum_{n=1}^{m} p(\mathbf{o}|\hat{\mathbf{z}}_n)$ and $p(\mathbf{o}|\hat{\mathbf{z}}_n) =$ softmax$\{\mathbf{W}_u \cdot \hat{\mathbf{z}}_n/\tau\}[\mathbf{o}]$. The weighting reflects how reliably the selected instruction embeddings as visual evidence support the possible existence of the object in the input image associated with the object token $\mathbf{o}$. When $\mathbf{o}$ is hallucinated, this weight tends to be low, thus reducing the consistency score.

Finally, the InsLen score $S_{\text{Ins}}(\mathbf{o})$ is proposed to integrate the local visual evidence from the Calibrated Local Score $S_{cls}$ with the context consistency measured by the Context Consistency Score $S_{ccs}$, enabling more reliable object hallucination detection. Formally, the overall scoring function is defined as a weighted combination of $S_{cls}$ and $S_{ccs}$, i.e.,

$$S_{\text{Ins}}(\mathbf{o}) = \omega \cdot S_{cls}(\mathbf{o}) + (1-\omega) \cdot S_{ccs}(\mathbf{o}), \quad (8)$$

where hyperparameter $\omega \in [0,1]$. The InsLen score provides confidence for the object token to perform object hallucination detection defined in Section 3.

## 5. Experiments

### 5.1. Experimental Setups

**Datasets** **MSCOCO** (Lin et al., 2014) dataset and **Object365** (Shao et al., 2019) dataset are used as the evaluation benchmarks. MSCOCO contains 80 object categories, while

Objects365 includes 365 object categories. During evaluation, the input instruction $\mathbf{p}_o$ is fixed as "*Please describe the image in detail.*". The **POPE** benchmark (Li et al., 2023) is used to evaluate the OH detection performance beyond object words, which formulates a binary classification task by instructing MLLMs with the question "*Is there a {object class} in this image?*" to answer "yes" or "no". To evaluate hallucinations beyond object existence, we further adopt the **CLEVR** (Johnson et al., 2017) dataset, which contains questions involving object attributes and relational reasoning, such as color, material, and relative spatial positions.

**Models** LLaVA-1.5-7B (Liu et al., 2024b) and InstructBLIP-7B (Dai et al., 2023) are selected as the representative open-source MLLMs for evaluation. Given that recent open-source LVLMs have incorporated more advanced techniques, such as DeepStack (Meng et al., 2024), we further include several newly released models in our evaluation, including mPLUG-Owl3-8B (Ye et al., 2025), Qwen3-VL-8B (Bai et al., 2025), and LLaVA-OneVision-Instruct-8B (An et al., 2025). This broader selection enables us to examine the robustness and generalizability of our method across diverse model architectures.

**Baselines** The baselines include general hallucination detection methods as well as recently proposed approaches specifically designed for OH detection. For methods based on

*Table 1.* Performance comparison with object hallucination detection methods on MSCOCO and Objects365 benchmarks. **Best** and second-best results are indicated in bold and underline, respectively. Appendix Table 9 reports the standard deviations.

| Method | Venue | LLaVA-1.5-7B | | InstructBLIP-7B | | mPLUG-Owl3-8B | | LLaVA-OneVision1.5-8B | | Qwen3-VL-8B | |
|---|---|---|---|---|---|---|---|---|---|---|---|
| | | AUROC ↑ | AUPR ↑ | AUROC ↑ | AUPR ↑ | AUROC ↑ | AUPR ↑ | AUROC ↑ | AUPR ↑ | AUROC ↑ | AUPR ↑ |
| | | *MSCOCO benchmark* | | | | | | | | | |
| NLL | ICLR'21 | 65.19 | 84.15 | 78.41 | 90.57 | 66.14 | 93.27 | 64.33 | 87.20 | 64.12 | 91.49 |
| Entropy | ICLR'24 | 63.29 | 82.21 | 67.78 | 80.73 | 57.32 | 88.83 | 65.56 | 86.79 | 66.23 | 91.89 |
| Internal Conf. | ICLR'25 | 73.45 | 91.21 | 81.62 | 93.27 | 65.07 | 94.40 | 71.89 | 93.07 | 67.68 | 92.03 |
| SVAR | CVPR'25 | 73.97 | 91.33 | 77.59 | 92.63 | 70.36 | 95.66 | 73.08 | 95.13 | 75.36 | 93.44 |
| Contextual Lens | NAACL'25 | 70.83 | 87.88 | 73.91 | 93.04 | 57.55 | 91.61 | 61.02 | 92.61 | 63.35 | 88.43 |
| EASY | ICCV'25 | 68.05 | 85.29 | 65.81 | 79.00 | 66.30 | 92.43 | 70.43 | 93.37 | 71.02 | 92.89 |
| GLSIM | NeurIPS'25 | 83.24 | 93.81 | 82.38 | 93.57 | 67.12 | 94.31 | 68.03 | 93.24 | 73.32 | 93.41 |
| InsLen (Ours) | | **86.93** | **96.38** | **85.50** | **94.74** | **75.11** | **96.47** | **82.41** | **96.99** | **81.02** | **96.70** |
| | | *Objects365 benchmark* | | | | | | | | | |
| NLL | ICLR'21 | 61.82 | 68.79 | 61.56 | 76.69 | 55.89 | 72.17 | 57.95 | 71.99 | 64.52 | 76.93 |
| Entropy | ICLR'24 | 58.74 | 63.05 | 57.96 | 62.80 | 50.86 | 70.66 | 57.06 | 74.80 | 65.46 | 67.44 |
| Internal Conf. | ICLR'25 | 70.45 | 76.78 | 71.46 | 79.35 | 64.57 | 78.55 | 64.10 | 79.95 | 64.50 | 76.90 |
| SVAR | CVPR'25 | 68.18 | 76.82 | 69.71 | 80.14 | 64.37 | 69.98 | 67.86 | 80.13 | 70.84 | 78.63 |
| Contextual Lens | NAACL'25 | 60.22 | 65.18 | 65.14 | 73.70 | 54.62 | 70.21 | 55.31 | 68.70 | 64.19 | 67.44 |
| EASY | ICCV'25 | 61.40 | 71.55 | 64.46 | 67.17 | 63.33 | 70.63 | 66.22 | 75.99 | 68.30 | 74.56 |
| GLSIM | NeurIPS'25 | 72.16 | 76.04 | 68.19 | 75.55 | 60.21 | 69.60 | 67.56 | 70.91 | 69.33 | 72.61 |
| InsLen (Ours) | | **76.32** | **78.49** | **76.18** | **83.70** | **72.43** | **79.90** | **72.66** | **81.73** | **77.44** | **80.12** |

*Table 2.* Performance comparison on the POPE benchmark. The reported results are averaged over three sampling strategies provided by POPE. Detailed results for each sampling strategy are presented in Appendix Table 11.

| Method | LLaVA-1.5-7B | | InstructBLIP-7B | | mPLUG-Owl3-8B | | LLaVA-OneVision1.5-8B | | Qwen3-VL-8B | |
|---|---|---|---|---|---|---|---|---|---|---|
| | AUROC ↑ | AUPR ↑ | AUROC ↑ | AUPR ↑ | AUROC ↑ | AUPR ↑ | AUROC ↑ | AUPR ↑ | AUROC ↑ | AUPR ↑ |
| NLL | 64.26 | 88.84 | 53.20 | 86.64 | 59.51 | 88.46 | 68.43 | 92.27 | 55.19 | 92.59 |
| Entropy | 52.59 | 85.66 | 54.65 | 87.55 | 55.35 | 87.35 | 60.73 | 84.69 | 73.52 | 92.89 |
| Internal Conf. | 54.41 | 85.87 | 52.00 | 81.77 | 55.46 | 89.63 | 63.74 | 86.01 | 46.39 | 87.54 |
| SVAR | 51.62 | 85.21 | 58.07 | 87.18 | 67.16 | 90.02 | 63.62 | 85.00 | 61.47 | 88.23 |
| Contextual Lens | 53.46 | 83.17 | 56.07 | 83.94 | 46.64 | 86.75 | 54.25 | 87.35 | 61.42 | 86.97 |
| GLSIM | 70.13 | 87.32 | 58.20 | 88.46 | 68.38 | 89.40 | 60.60 | 87.89 | 68.24 | 88.44 |
| InsLen (Ours) | **83.94** | **96.14** | **74.09** | **94.59** | **79.42** | **96.45** | **76.50** | **95.90** | **75.42** | **93.70** |

token-level logit outputs, we consider Entropy (Malinin & Gales, 2021), NLL (Negative Log-Likelihood) (Zhou et al., 2024), and Internal Conf (Jiang et al., 2025a). Attention-based methods include SVAR (Jiang et al., 2025b) and EAZY (Che et al., 2025), which leverage attention scores to identify hallucinated objects. For hidden state based methods, we adopt GLSIM (Park & Li, 2025) and ContextLens (Phukan et al., 2025).

**Implementation Details** In our experiments, instruction embeddings $\{z_j\}_{j=1}^M$ and each object token embedding $h_o$ in the answer are extracted from the penultimate layer of the language model (e.g., the 31$^{nd}$ layer for LLaVA and the 35$^{th}$ layer for Qwen3-VL). Unless otherwise specified, the hyperparameters are set to $\omega = 0.4$, $\alpha = 2$, $\tau = 10$, and the number $m$ of selected instruction embeddings is 4. Hyperparameters for the vision-based scores follow the original settings. The maximum generation length is set to 512 tokens. All the models and detectors are implemented on NVIDIA DRIVE-PG199-RPOD GPUs with 32 GB memory

each. Please refer to Appendix A.3 for more details.

**Evaluation** As stated in Section 3, we formulate object hallucination detection as a binary classification task, where the scoring function provides the confidence for each object token. For MSCOCO, Object365, and CLEVR benchmarks, we randomly sample 5,000 images for each experiment and report the average results over three random seeds. Following CHAIR (Rohrbach et al., 2018), we extract object tokens from the generated descriptions and match them with the ground-truth object labels and their corresponding synonyms for each image to determine whether an object token is hallucinated. For object names consisting of multiple tokens, we follow prior work (Park & Li, 2025; Jiang et al., 2025b) and use only the first token for detection. Following prior work (Park & Li, 2025), we adopt AUROC (area under the receiver operating characteristic curve) and AUPR (area under the precision-recall curve) as evaluation metrics.

*Table 3.* Performance comparison on the CLEVR benchmark. Appendix Table 10 reports the corresponding standard deviations.

| Method | LLaVA-1.5-7B | | InstructBLIP-7B | | mPLUG-Owl3-8B | | LLaVA-OneVision1.5-8B | | Qwen3-VL-8B | |
|---|---|---|---|---|---|---|---|---|---|---|
| | AUROC ↑ | AUPR ↑ | AUROC ↑ | AUPR ↑ | AUROC ↑ | AUPR ↑ | AUROC ↑ | AUPR ↑ | AUROC ↑ | AUPR ↑ |
| NLL | 50.58 | 46.02 | 49.39 | 32.66 | 64.34 | 93.25 | 63.90 | 99.61 | 58.25 | 97.11 |
| Entropy | 51.92 | 48.21 | 60.93 | 50.42 | 63.01 | 91.75 | 64.00 | 98.73 | 72.67 | 97.11 |
| Internal Conf. | 41.72 | 38.72 | 47.18 | 35.14 | 53.76 | 90.28 | 51.79 | 99.40 | 56.11 | 97.92 |
| SVAR | 50.82 | 44.22 | **62.09** | **50.95** | 70.71 | 93.25 | 51.23 | 99.46 | 73.01 | 98.61 |
| Contextual Lens | 49.32 | 45.26 | 47.68 | 35.14 | 65.62 | 93.36 | 52.39 | 99.48 | 56.89 | 97.88 |
| GLSIM | 50.31 | 49.81 | 48.61 | 36.88 | 68.47 | 94.63 | 70.26 | 99.52 | 62.71 | 97.95 |
| InsLen(Ours) | **62.34** | **55.37** | 61.32 | 50.76 | **74.01** | **95.74** | **75.52** | **99.84** | **77.72** | **98.95** |

## 5.2. Main results

**Evaluation on OH Detection Benchmarks.** As shown in Table 1, the proposed InsLen score achieves the best OH detection performance on both the MSCOCO and Objects365 benchmarks across different MLLMs. For example, on Qwen3-VL, InsLen outperforms the strongest baseline GLSIM on the MSCOCO benchmark by 7.70% in AUROC and 3.29% in AUPR. It also surpasses SVAR on the Object365 benchmark by 6.60% in AUROC and 1.49% in AUPR. These results demonstrate that our method generalizes well across different model architectures. On the POPE benchmark, our method also achieves the best performance in all settings. As shown in Table 2, InsLen surpasses the strongest baseline GLSIM by up to 13.81% in AUROC for LLaVA-1.5 and consistently outperforms all baselines. This demonstrates that, by selecting the instruction embeddings with high confidence assigned to the detected tokens, InsLen can also effectively adapt to the POPE setting. Overall, these results support that InsLen maintains robust OH detection capability under different task formulations.

**Evaluation on the CLEVR Benchmark.** The CLEVR dataset enables the assessment of attribute-level and relational hallucinations. As shown in Table 3, although LLaVA and InstructBLIP show limited answer accuracy on this benchmark, our method still achieves the best and second-best results. When deployed on stronger models such as mPLUG-Owl3 and Qwen3-VL, our method outperforms other baselines by a clear margin. For example, it achieves an AUROC of 77.72% on Qwen3-VL, compared to 73.01% for SVAR. These results suggest that our InsLen score enables more effective detection for attributes and relations.

## 5.3. Ablation and Sensitivity Studies

**Ablation Study of the InsLen Score.** As shown in Table 4, we conduct an ablation study on the individual components of InsLen. For the Calibrated Local Score which consists of the vision-base score $S_{local}$ and the Calibration Confidence $S_{cafe}$, incorporating $S_{cafe}$ brings an AUROC improvement of 10.11% over using $S_{local}$ alone on LLaVA-1.5 (row 1 vs. row 3). For the Context Consistency Score, combining the

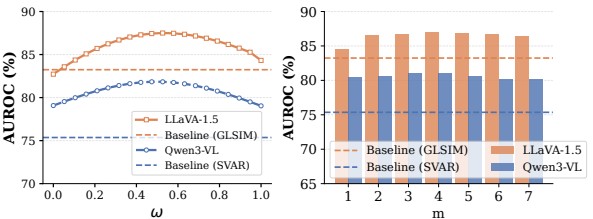

*Figure 5.* Sensitivity studies of hyperparameter $\omega$ and the number $m$ of selected instruction embeddings for InsLen score on the MSCOCO dataset. Dashed lines indicate the strongest baselines for LLaVA-1.5 and Qwen3-VL, respectively.

*Table 4.* Ablation study of the components in InsLen score, where the MSCOCO dataset is used. 'Conf.' denote the averaged confidence (Eq. 7).

| $S_{local}$ | $S_{cafe}$ | $S_{con}$ | Conf. | LLaVA-1.5 | | Qwen3-VL | |
|---|---|---|---|---|---|---|---|
| | | | | AUROC↑ | AUPR↑ | AUROC↑ | AUPR↑ |
| ✓ | - | - | - | 74.20 | 89.47 | 65.43 | 87.85 |
| - | ✓ | - | - | 80.41 | 93.59 | 77.06 | 95.74 |
| ✓ | ✓ | - | - | 84.31 | 94.37 | 79.83 | 95.44 |
| - | - | ✓ | - | 80.69 | 93.76 | 71.94 | 90.26 |
| - | - | - | ✓ | 79.44 | 93.33 | 78.12 | 94.43 |
| - | - | ✓ | ✓ | 82.72 | 93.83 | 79.07 | 95.06 |
| ✓ | - | ✓ | - | 84.46 | 95.31 | 79.55 | 95.14 |
| ✓ | ✓ | ✓ | - | 84.07 | 94.86 | 79.77 | 95.27 |
| ✓ | - | ✓ | ✓ | 85.46 | 95.43 | 80.01 | 95.02 |
| ✓ | ✓ | ✓ | ✓ | **86.93** | **96.38** | **81.02** | **96.70** |

consistency score $S_{con}$ with the averaged confidence (Conf.) leads to AUROC gains of 2.03% on LLaVA-1.5 and 7.23% on Qwen3-VL (row 4 vs. row 6), respectively, supporting that confidence cues can further enhance consistency-based detection. Moreover, the efficacy of different design variants of $S_{ccs}$ is provided in Appendix A.2. When all components are integrated into the full InsLen score (last row), the performance is further improved to the best AUROC of 86.93% on LLaVA-1.5 and 81.02% on Qwen3-VL, suggesting that each proposed component contributes positively to object hallucination detection. Moreover, the consistent improvement obtained by integrating the Context Consistency Score with the Calibrated Local Score suggests that the former captures object-level contextual cues complementary to the

*Table 5.* Object hallucination detection performance when combining the proposed Cafe with different vision-based methods. The MSCOCO benchmark is used.

| Method | Cafe | LLaVA-1.5 | | Qwen3-VL | |
|---|---|---|---|---|---|
| | | AUROC | AUPR | AUROC | AUPR |
| Internal Conf. | ✗ | 73.45 | 91.21 | 67.68 | 92.03 |
| | ✓ | **83.93** | **94.39** | **78.84** | **96.06** |
| LSS | ✗ | 74.20 | 89.47 | 65.43 | 87.85 |
| | ✓ | **84.31** | **94.37** | **77.16** | **95.77** |
| SVAR | ✗ | 73.97 | 91.33 | 75.36 | 93.44 |
| | ✓ | **81.44** | **94.42** | **80.21** | **95.97** |

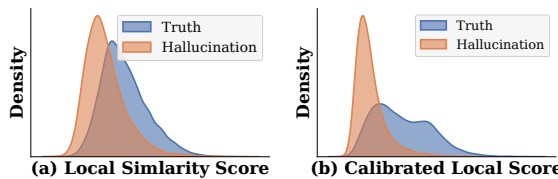

*Figure 6.* Comparison of the Local Similarity Score (LSS) distributions before and after calibration for LLaVA-1.5 on MSCOCO.

*Table 6.* Performance comparison under different input instructions on MSCOCO. Results are averaged within each instruction type. HR (%) represents the hallucination rate, i.e., the fraction of hallucinated object tokens over all generated object tokens.

| Instruction type | LLaVA-1.5 | | | Qwen3-VL | | |
|---|---|---|---|---|---|---|
| | HR | AUROC | AUPR | HR | AUROC | AUPR |
| Original instruction | 22.34 | 86.93 | **96.38** | 10.66 | 81.02 | **96.70** |
| Long instruction | 25.51 | **89.33** | 95.46 | 10.10 | **81.33** | 96.21 |
| Short instruction | 21.90 | 85.63 | 94.86 | 10.54 | 80.72 | 95.33 |

patch-level visual evidence modeled by the latter, resulting in more robust object hallucination detection.

**Sensitivity of Balance Coefficient $\omega$.** We conduct the sensitivity study on the balance factor $\omega$. As shown in Figure 5 (left), across a wide range of $\omega \in [0.1, 0.9]$, the combined score consistently outperforms the best baseline (dashed lines), indicating that the performance of our method is insensitive to the choice of $\omega$. Moreover, the performance peaks around $\omega = 0.6$, suggesting that the two components contribute comparably and are well balanced.

**Sensitivity of the Number of Instruction Embeddings.** We examine the impact of the number $m$ of instruction embeddings used in the Context Consistency Score. As shown in Figure 5 (right), the OH detection performance remains stably well when $m$ is in the range $[2, 5]$, higher than the best baselines (dashed lines), suggesting that aggregating sufficient object-related information from instruction embeddings facilitates more accurate hallucination detection. See more sensitivity studies in Appendix A.2.

## 5.4. Further Analysis

**Generalization across Vision-based Scores.** The Calibration Confidence (Cafe) serves as a calibration factor that adjusts the visual evidence provided by existing vision-based detectors. Beyond the Local Similarity Score (LSS) used in InsLen, we further evaluate the effect of Cafe when combined with other vision-based scores. As demonstrated in Table 5, combining Cafe with various vision-based scores consistently improves the baseline detection performance. For example, when combining SVAR with Cafe, the Calibrated Local Score improves AUROC by 7.47%. These results confirm that Cafe can effectively calibrate visual confidence and enhance the separability between hallucinated and real objects, which is also illustrated in Figure 6.

**Impact of the Length of the Input Instruction.** To examine the impact of input instructions on the InsLen score, we design three types of instructions with different lengths but identical semantic content (see Appendix Table 14 for de-

tails). As shown in Table 6, for LLaVA, our method exhibits improved hallucination detection performance with a longer input instructions. Compared to the original instruction, the AUROC increases by 2.40% when longer instruction is used. This improvement is likely due to the increased number of instruction embeddings induced by the longer instruction, which allows the model to encode richer visual information within the instruction embeddings. On the other hand, for Qwen3-VL, our method demonstrates relatively stable performance across different instructions, although a slight improvement is observed under the longer instruction. These results suggest that more detailed and informative instructions are often beneficial for hallucination detection.

**Performance on Post-trained Models.** Recent approaches apply post-training techniques to better align MLLMs with human preferences. To examine the efficacy of our method on post-trained models, we evaluate our method on two post-trained LLaVA variants: LLaVA-RLHF-7B (Sun et al., 2024) and LLaVA-RLAIF-V-7B (Yu et al., 2025). As shown in Table 7, our InsLen score achieves the best performance on both models. Notably, LLaVA-RLAIF-V exhibits a substantially lower hallucination rate (HR = 6.72%), indicating that many easy hallucination cases are already mitigated by post-training, and more challenging object hallucinations persist during the testing. Under this challenging evaluation setting, InsLen still outperforms all competing methods. Overall, these results support that InsLen remains effective on post-trained models.

**Computational Overhead.** Since our InsLen introduces extra operations, it incurs slightly higher computational overhead compared to several baselines, as shown in Table 8. However, the cost is minimal when compared to

*Table 7.* Performance comparison on post-trained LLaVA variants on the MSCOCO benchmark.

| Method | LLaVA-RLHF-7B | | LLaVA-RLAIF-V-7B | |
|---|---|---|---|---|
| | AUROC | AUPR | AUROC | AUPR |
| GLSIM | 83.68 | 94.79 | 72.36 | 96.89 |
| Internal Conf. | 71.23 | 90.33 | 67.76 | 96.48 |
| InsLen | **86.37** | **95.66** | **80.14** | **98.18** |

*Table 8.* Latency comparison between LLaVA and Qwen3-VL

| Operation | LLaVA (ms) | Qwen3-VL(ms) |
|---|---|---|
| Answer generation | 1870.0 | 19550.0 |
| GLSIM | 95.2 | 537.2 |
| SVAR | 12.9 | 15.7 |
| EASY | 3896.7 | 40293.3 |
| InsLen (Ours) | 104.5 | 564.5 |
| -Projecting image embeddings | 38.5 | 97.6 |
| -Projecting instruction embeddings | 4.5 | 15.5 |

answer generation, and thus still meets the requirements for inference-time detection.

**Limitations.** The use of Logit Lens enables us to translate the visual information in instructions into text form. However, the visual information stored in the instruction features by the model does not necessarily align with the generated answers, but is encoded as synonyms, as illustrated in Appendix Figure 14-16. Therefore, selecting confidence scores based on the generated answer may not fully capture all the semantic information contained in the instruction features. Moreover, potential issues noted in previous work (Belrose et al., 2023), such as representation drift, may also affect our method. In Appendix Section A.6, we further discuss the impact of model architecture on performance.

## 6. Conclusion

This study reveals that instruction embeddings in multimodal large language models implicitly encode reliable visual semantics and can effectively suppress misleading visual information. Based on this observation, we propose the Instruction Lens Score, a training-free and plug-and-play object hallucination detector that combines calibrated visual evidence with instruction-based context consistency. Extensive experiments across multiple benchmarks and diverse MLLM architectures demonstrate that InsLen consistently outperforms existing methods. Our findings suggest that instruction embeddings provide a valuable and underexplored signal for improving the reliability of MLLM generation.

## Impact Statement

This paper presents work whose goal is to advance the field of Machine Learning. There are many potential societal consequences of our work, none of which we feel must be specifically highlighted here.

## Acknowledgements

This work is supported in part by the National Natural Science Foundation of China (Grant No. 62571559), the Guangdong S&T Program (Grant No. 2024B0101040005), the Major Key Project of PCL (Grant No. PCL2025AS209), and Guangdong Excellent Youth Team Program (Grant No.2023B1515040025).

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

# A. More Analysis

## A.1. More experiment results.

**Results on MSCOCO, Object365, and CLEVR.** As shown in Tables 9 and 10, we report the detailed performance of each method with the corresponding standard deviation to reflect performance stability. In addition, we also provide the individual performance of the Context Consistency Score and the Calibrated Local Score.

*Table 9.* Performance comparison on MSCOCO and Objects365 benchmarks. CCS and CLS denote the Context Consistency Score and the Calibrated Local Score, respectively.

| Method | LLaVA-1.5-7B | | InstructBLIP-7B | | mPLUG-Owl3-8B | | LLaVA-OneVision1.5-8B | | Qwen3-VL-8B | |
|---|---|---|---|---|---|---|---|---|---|---|
| | AUROC ↑ | AUPR ↑ | AUROC ↑ | AUPR ↑ | AUROC ↑ | AUPR ↑ | AUROC ↑ | AUPR ↑ | AUROC ↑ | AUPR ↑ |
| *MSCOCO benchmark* | | | | | | | | | | |
| NLL | $65.19_{\pm1.42}$ | $84.15_{\pm1.11}$ | $78.41_{\pm1.76}$ | $90.57_{\pm1.48}$ | $66.14_{\pm1.95}$ | $93.27_{\pm1.38}$ | $64.33_{\pm2.63}$ | $87.20_{\pm1.84}$ | $64.12_{\pm1.57}$ | $91.49_{\pm1.05}$ |
| Entropy | $63.29_{\pm2.21}$ | $82.21_{\pm1.66}$ | $67.78_{\pm2.73}$ | $80.73_{\pm1.92}$ | $57.32_{\pm2.14}$ | $88.83_{\pm1.44}$ | $65.56_{\pm1.89}$ | $86.79_{\pm1.37}$ | $66.23_{\pm1.58}$ | $91.89_{\pm1.26}$ |
| Internal Conf. | $73.45_{\pm1.35}$ | $91.21_{\pm1.14}$ | $81.62_{\pm1.82}$ | $93.27_{\pm1.27}$ | $65.07_{\pm2.41}$ | $94.40_{\pm1.52}$ | $71.89_{\pm1.76}$ | $93.07_{\pm1.69}$ | $67.68_{\pm2.08}$ | $92.03_{\pm1.93}$ |
| SVAR | $73.97_{\pm1.44}$ | $91.33_{\pm1.28}$ | $77.59_{\pm2.05}$ | $92.63_{\pm1.39}$ | $70.36_{\pm1.87}$ | $95.66_{\pm1.22}$ | $73.08_{\pm1.69}$ | $95.13_{\pm1.31}$ | $75.36_{\pm1.53}$ | $93.44_{\pm1.47}$ |
| Contextual Lens | $70.83_{\pm2.24}$ | $87.88_{\pm1.95}$ | $73.91_{\pm1.68}$ | $93.04_{\pm1.42}$ | $57.55_{\pm2.81}$ | $91.61_{\pm1.77}$ | $61.02_{\pm2.36}$ | $92.61_{\pm1.63}$ | $63.35_{\pm2.12}$ | $88.43_{\pm1.84}$ |
| EASY | $68.05_{\pm2.47}$ | $85.29_{\pm2.01}$ | $65.81_{\pm2.66}$ | $79.00_{\pm2.59}$ | $66.30_{\pm2.18}$ | $92.43_{\pm1.51}$ | $70.43_{\pm1.73}$ | $93.37_{\pm1.44}$ | $71.02_{\pm1.69}$ | $92.89_{\pm1.48}$ |
| GLSIM | $83.24_{\pm1.52}$ | $93.81_{\pm1.31}$ | $82.38_{\pm1.37}$ | $93.57_{\pm1.29}$ | $67.12_{\pm2.09}$ | $94.31_{\pm1.34}$ | $68.03_{\pm2.14}$ | $93.24_{\pm1.41}$ | $73.32_{\pm1.85}$ | $93.41_{\pm1.36}$ |
| CLS (Ours) | $84.31_{\pm1.28}$ | $94.37_{\pm1.34}$ | $83.09_{\pm1.47}$ | $94.00_{\pm1.12}$ | $68.54_{\pm2.63}$ | $94.21_{\pm1.39}$ | $80.62_{\pm1.71}$ | $96.17_{\pm1.18}$ | $79.83_{\pm1.86}$ | $95.44_{\pm1.25}$ |
| CCS (Ours) | $82.72_{\pm1.36}$ | $93.83_{\pm1.42}$ | $76.18_{\pm2.11}$ | $88.39_{\pm1.95}$ | $72.43_{\pm2.58}$ | $95.66_{\pm1.31}$ | $81.66_{\pm1.47}$ | $96.48_{\pm1.18}$ | $79.07_{\pm1.69}$ | $95.06_{\pm1.27}$ |
| InsLen (Ours) | $86.93_{\pm1.18}$ | $96.38_{\pm1.11}$ | $85.50_{\pm1.26}$ | $94.74_{\pm1.19}$ | $75.11_{\pm1.67}$ | $96.47_{\pm1.05}$ | $82.41_{\pm1.33}$ | $96.99_{\pm1.02}$ | $81.02_{\pm1.41}$ | $96.70_{\pm1.07}$ |
| *Objects365 benchmark* | | | | | | | | | | |
| NLL | $61.82_{\pm3.42}$ | $68.79_{\pm2.95}$ | $69.56_{\pm3.11}$ | $76.69_{\pm2.64}$ | $55.89_{\pm4.26}$ | $72.17_{\pm3.02}$ | $57.95_{\pm3.88}$ | $71.99_{\pm2.91}$ | $64.52_{\pm2.74}$ | $76.93_{\pm2.48}$ |
| Entropy | $58.74_{\pm4.91}$ | $63.05_{\pm4.22}$ | $57.96_{\pm4.03}$ | $62.80_{\pm3.15}$ | $50.86_{\pm3.76}$ | $70.66_{\pm2.84}$ | $57.06_{\pm3.12}$ | $74.80_{\pm2.51}$ | $65.46_{\pm2.89}$ | $67.44_{\pm3.07}$ |
| Internal Conf. | $70.45_{\pm2.67}$ | $76.78_{\pm2.41}$ | $71.46_{\pm2.54}$ | $79.35_{\pm2.18}$ | $64.57_{\pm3.21}$ | $78.55_{\pm2.36}$ | $64.10_{\pm3.34}$ | $79.95_{\pm2.09}$ | $64.50_{\pm3.12}$ | $76.90_{\pm2.45}$ |
| SVAR | $68.18_{\pm2.88}$ | $76.82_{\pm2.39}$ | $69.71_{\pm2.76}$ | $80.14_{\pm2.07}$ | $64.37_{\pm3.29}$ | $69.98_{\pm2.81}$ | $67.86_{\pm2.93}$ | $80.13_{\pm2.05}$ | $70.84_{\pm2.61}$ | $78.63_{\pm2.22}$ |
| Contextual Lens | $60.22_{\pm3.47}$ | $65.18_{\pm3.06}$ | $65.14_{\pm3.11}$ | $73.70_{\pm2.52}$ | $54.62_{\pm4.08}$ | $70.21_{\pm2.83}$ | $55.31_{\pm3.96}$ | $68.70_{\pm2.91}$ | $64.19_{\pm3.18}$ | $67.44_{\pm3.05}$ |
| EASY | $61.40_{\pm3.26}$ | $71.55_{\pm2.61}$ | $64.46_{\pm3.09}$ | $67.17_{\pm3.02}$ | $63.33_{\pm3.18}$ | $70.63_{\pm2.77}$ | $66.22_{\pm2.85}$ | $75.99_{\pm2.34}$ | $68.30_{\pm2.71}$ | $74.56_{\pm2.49}$ |
| GLSIM | $72.16_{\pm2.58}$ | $76.04_{\pm2.42}$ | $68.19_{\pm2.83}$ | $75.55_{\pm2.51}$ | $60.21_{\pm3.49}$ | $69.60_{\pm2.88}$ | $67.56_{\pm2.91}$ | $70.91_{\pm2.76}$ | $69.33_{\pm2.69}$ | $72.61_{\pm2.63}$ |
| CLS (Ours) | $75.49_{\pm2.83}$ | $78.48_{\pm2.47}$ | $75.67_{\pm2.65}$ | $81.79_{\pm2.18}$ | $71.28_{\pm3.74}$ | $76.36_{\pm2.59}$ | $71.78_{\pm3.41}$ | $82.94_{\pm2.03}$ | $74.26_{\pm2.91}$ | $79.21_{\pm2.36}$ |
| CCS (Ours) | $73.70_{\pm1.84}$ | $79.62_{\pm2.13}$ | $70.62_{\pm1.57}$ | $79.56_{\pm2.08}$ | $69.22_{\pm2.41}$ | $80.95_{\pm1.69}$ | $70.36_{\pm1.93}$ | $77.83_{\pm2.27}$ | $70.83_{\pm1.76}$ | $74.44_{\pm2.54}$ |
| InsLen (Ours) | $76.32_{\pm2.37}$ | $78.49_{\pm2.26}$ | $76.18_{\pm2.41}$ | $83.70_{\pm2.03}$ | $72.43_{\pm2.78}$ | $79.90_{\pm2.21}$ | $72.66_{\pm2.69}$ | $81.73_{\pm2.11}$ | $77.44_{\pm2.33}$ | $80.12_{\pm2.19}$ |

**Results on POPE benchmark.** The POPE benchmark supports multiple object sampling strategies, including random sampling, popular sampling, and adversarial sampling. We provide the results on each sampling strategy in Table 11.

**Impact of model scaling.** As shown in Table 12, we evaluate InsLen on larger-scale models, including LLaVA-1.5-13B and InstructBLIP-13B. InsLen consistently outperforms the strongest baselines, indicating that its effectiveness generalizes well to larger model sizes.

## A.2. More ablation and sensitivity studies.

**Impact of instruction embedding layer $l$.** We conduct a sensitivity study on the instruction embedding layer $l$ for the proposed InsLen score, as the choice of $l$ directly affects the computation of both the Calibration Confidence in Eq. (4) and the Context Consistency Score in Eq. (6). As shown in Figure 7, for the LLaVA model, using deeper instruction embeddings leads to better OH detection performance. This suggests that higher-layer instruction representations are more informative for identifying reliable visual evidence. In contrast, Qwen3-VL shows stronger sensitivity to the layer choice: although deeper layers generally perform better, the performance varies noticeably across layers. We attribute this behavior to the intermediate fusion mechanism (i.e., DeepStack) in Qwen3-VL, which integrates visual encoder features into the language model and may disrupt the progressive abstraction of visual semantics across layers. Overall, deeper instruction embeddings are generally more effective for hallucination detection, while architectural differences can affect the stability of this trend.

**Sensitivity studies of temperature hyperparameter $\tau$ and $\alpha$.** As shown in the left two subfigures of Figure 8, as the temperature hyperparameter $\tau$ increases, the model performance first improves and then gradually plateaus. Higher $\tau$ smooths the distribution, reducing overconfident predictions and thereby improving the separability between hallucinated and real object tokens. Overall, our method is robust to the choice of $\tau$, achieving consistently strong performance across a broad range of $\tau$ values. As shown in the middle two subfigures of Figure 8, our method shows stable performance with respect to the hyperparameter $\alpha$ over a moderate range of $[0, 5]$, and achieves slightly better results when $\alpha$ is around 2.

*Table 10.* Performance comparison on the CLEVR benchmark.

| Method | LLaVA-1.5-7B | | InstructBLIP-7B | | mPLUG-Owl3-8B | | LLaVA-OneVision1.5-8B | | Qwen3-VL-8B | |
|---|---|---|---|---|---|---|---|---|---|---|
| | AUROC ↑ | AUPR ↑ | AUROC ↑ | AUPR ↑ | AUROC ↑ | AUPR ↑ | AUROC ↑ | AUPR ↑ | AUROC ↑ | AUPR ↑ |
| NLL | $50.58_{\pm1.27}$ | $46.02_{\pm2.91}$ | $49.39_{\pm0.84}$ | $32.66_{\pm3.45}$ | $64.34_{\pm1.96}$ | $93.25_{\pm0.73}$ | $63.90_{\pm2.38}$ | $99.61_{\pm0.12}$ | $58.25_{\pm3.11}$ | $97.11_{\pm1.08}$ |
| Entropy | $51.92_{\pm0.69}$ | $48.21_{\pm2.44}$ | $60.93_{\pm1.37}$ | $50.42_{\pm3.62}$ | $63.01_{\pm0.95}$ | $91.75_{\pm2.58}$ | $64.00_{\pm1.14}$ | $98.73_{\pm0.61}$ | $\underline{72.67}_{\pm2.07}$ | $97.11_{\pm1.83}$ |
| Internal Conf. | $41.72_{\pm3.26}$ | $38.72_{\pm1.91}$ | $47.18_{\pm2.75}$ | $35.14_{\pm0.88}$ | $53.76_{\pm1.63}$ | $90.28_{\pm2.41}$ | $51.79_{\pm3.02}$ | $99.40_{\pm0.49}$ | $56.11_{\pm2.54}$ | $97.92_{\pm1.36}$ |
| SVAR | $50.82_{\pm1.58}$ | $44.22_{\pm3.11}$ | $62.09_{\pm0.94}$ | $50.95_{\pm2.07}$ | $70.71_{\pm1.02}$ | $93.25_{\pm2.66}$ | $51.23_{\pm3.74}$ | $99.46_{\pm0.57}$ | $73.01_{\pm1.45}$ | $98.61_{\pm0.83}$ |
| Contextual Lens | $49.32_{\pm2.88}$ | $45.26_{\pm1.04}$ | $47.68_{\pm3.41}$ | $35.14_{\pm0.96}$ | $65.62_{\pm2.19}$ | $93.36_{\pm1.57}$ | $52.39_{\pm3.08}$ | $99.48_{\pm0.46}$ | $56.89_{\pm2.63}$ | $97.88_{\pm1.22}$ |
| GLSIM | $50.31_{\pm1.12}$ | $49.81_{\pm0.77}$ | $48.61_{\pm2.59}$ | $36.88_{\pm1.35}$ | $68.47_{\pm0.86}$ | $94.63_{\pm1.04}$ | $70.26_{\pm0.93}$ | $99.52_{\pm0.21}$ | $62.71_{\pm1.98}$ | $97.95_{\pm0.69}$ |
| CLS (Ours) | $53.78_{\pm1.84}$ | $60.37_{\pm0.97}$ | $53.76_{\pm2.63}$ | $50.16_{\pm1.21}$ | $68.47_{\pm0.88}$ | $94.77_{\pm1.45}$ | $65.14_{\pm2.09}$ | $98.90_{\pm0.54}$ | $65.13_{\pm1.76}$ | $95.72_{\pm0.92}$ |
| CCS (Ours) | $61.75_{\pm1.12}$ | $52.48_{\pm2.37}$ | $60.73_{\pm0.95}$ | $50.44_{\pm1.68}$ | $71.97_{\pm0.74}$ | $95.13_{\pm1.26}$ | $74.33_{\pm1.59}$ | $99.01_{\pm0.18}$ | $75.75_{\pm1.03}$ | $97.55_{\pm0.81}$ |
| InsLen(Ours) | $62.34_{\pm1.12}$ | $55.37_{\pm2.03}$ | $61.32_{\pm0.97}$ | $50.76_{\pm1.45}$ | $74.01_{\pm2.11}$ | $95.74_{\pm0.88}$ | $75.52_{\pm1.36}$ | $99.84_{\pm0.11}$ | $77.72_{\pm1.09}$ | $98.95_{\pm0.73}$ |

These results indicate that keeping a positive value of the Context Consistency Score improves the combined score.

**Alternative designs for InsLen score.** In the Context Consistency Score, consistency is measured by the relative discrepancy (Relative) between the generated object embedding $\mathbf{h}_o$ and the aggregated instruction embedding $\bar{\mathbf{I}}$, defined as $\alpha - \frac{||\mathbf{h_o}-\bar{\mathbf{I}}||}{||\mathbf{h_o}||}$. We further explore the following alternative consistency measures: (1) **Cos**: cosine similarity between $\mathbf{h}_o$ and $\bar{\mathbf{I}}$, focusing solely on directional alignment; (2) **Distance**: distance bewtween $\mathbf{h_o}$ and $\hat{\mathbf{I}}$, defined as $\alpha - ||\mathbf{h_o} - \bar{\mathbf{I}}||$, measuring absolute discrepancy without scale normalization; (3) **Direction**: normalized Euclidean distance between unit vectors, defined as $\alpha - ||\frac{\mathbf{h_o}}{||\mathbf{h_o}||} - \frac{\bar{\mathbf{I}}}{||\bar{\mathbf{I}}||}||$, which measures directional discrepancy while discarding magnitude information. As shown in Table 13, the cosine-similarity-based score achieves performance comparable to the original design. In contrast, the Distance and Direction variants exhibit unstable performance across different models, indicating that ignoring either scale normalization or magnitude information leads to less reliable consistency estimation.

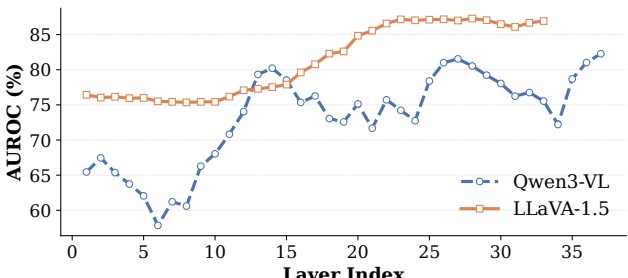

*Figure 7.* Sensitivity study of the decoder layer for instruction embedding extraction. Index '0' corresponds to the input embedding layer.

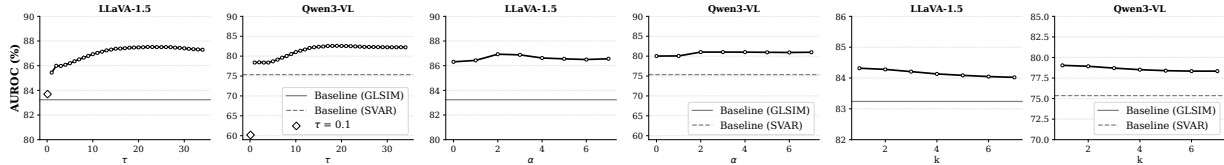

*Figure 8.* Sensitivity studies of temperature hyperparameters $\tau$, $\alpha$, and the top-$k$ choice in Calibration Confidence.

On the other hand, the Calibration Confidence used in the Calibrated Local Score is defined as the maximum probability assigned to the object token across all instruction embeddings. We further investigate an alternative design that uses the average confidence over the top-$k$ instruction embeddings with the highest probabilities for the object token. As shown in the right two subfigures of Figure 8, the performance of the Calibrated Local Score degrades slightly as $k$ increases, indicating that using the maximum confidence is more effective than averaging. This design is fundamentally different from

*Table 11.* Performance comparison on POPE benchmarks.

| Method | LLaVA-1.5-7B | | InstructBLIP-7B | | mPLUG-Owl3-8B | | LLaVA-OneVision1.5-8B | | Qwen3-VL-8B | |
|---|---|---|---|---|---|---|---|---|---|---|
| | AUROC ↑ | AUPR ↑ | AUROC ↑ | AUPR ↑ | AUROC ↑ | AUPR ↑ | AUROC ↑ | AUPR ↑ | AUROC ↑ | AUPR ↑ |
| *Popular Sampling* | | | | | | | | | | |
| NLL | 65.51 | 88.99 | 50.58 | 86.69 | 58.74 | 87.78 | 70.45 | 93.83 | 51.18 | 97.32 |
| Entropy | 50.19 | 85.70 | 52.35 | 86.07 | 56.35 | 85.83 | 62.37 | 82.25 | 74.61 | 91.42 |
| Internal Conf. | 54.83 | 86.88 | 53.56 | 81.50 | 54.07 | 89.50 | 60.30 | 85.65 | 45.45 | 88.29 |
| SVAR | 50.26 | 85.61 | 55.42 | 91.82 | 67.83 | 89.66 | 68.69 | 84.15 | 63.47 | 91.43 |
| Contextual Lens | 54.62 | 82.53 | 52.83 | 81.61 | 43.10 | 86.52 | 52.37 | 87.31 | 59.70 | 86.90 |
| GLSIM | 73.71 | 80.88 | 56.78 | 85.38 | 72.04 | 86.55 | 52.99 | 88.65 | 67.57 | 88.69 |
| CLS (Ours) | 62.75 | 82.96 | 54.42 | 82.22 | 61.17 | 90.97 | 69.68 | 94.71 | 67.83 | 93.90 |
| CCS (Ours) | 85.30 | 97.15 | 75.41 | 94.92 | 79.76 | 96.55 | 78.74 | 95.96 | 78.75 | 96.46 |
| InsLen (Ours) | 84.91 | 96.71 | 73.72 | 94.73 | 80.34 | 96.65 | 80.31 | 96.63 | 78.98 | 93.44 |
| *Adversarial Sampling* | | | | | | | | | | |
| NLL | 68.29 | 86.47 | 55.78 | 84.95 | 58.92 | 86.51 | 63.89 | 90.66 | 53.62 | 87.14 |
| Entropy | 50.86 | 80.32 | 55.57 | 84.27 | 53.17 | 85.66 | 63.22 | 83.33 | 68.43 | 90.07 |
| Internal Conf. | 55.94 | 81.65 | 51.42 | 75.54 | 49.93 | 85.90 | 63.21 | 84.95 | 51.49 | 84.80 |
| SVAR | 50.21 | 79.60 | 53.32 | 80.64 | 66.33 | 85.68 | 56.61 | 84.86 | 65.62 | 86.14 |
| Contextual Lens | 53.26 | 80.54 | 57.39 | 82.07 | 49.87 | 82.36 | 53.79 | 86.24 | 59.34 | 85.08 |
| GLSIM | 66.33 | 87.75 | 57.86 | 89.70 | 64.35 | 86.25 | 63.40 | 86.59 | 66.82 | 86.32 |
| CLS (Ours) | 59.17 | 82.96 | 52.33 | 83.43 | 57.95 | 87.82 | 65.79 | 93.06 | 65.93 | 91.99 |
| CCS (Ours) | 81.00 | 93.95 | 76.79 | 94.69 | 78.05 | 95.25 | 76.68 | 93.45 | 70.41 | 93.67 |
| InsLen (Ours) | 81.32 | 93.96 | 75.80 | 93.27 | 78.10 | 95.21 | 71.98 | 94.44 | 72.02 | 92.05 |
| *Random Sampling* | | | | | | | | | | |
| NLL | 58.99 | 91.07 | 53.25 | 88.28 | 60.86 | 91.09 | 70.96 | 92.33 | 60.78 | 93.32 |
| Entropy | 56.71 | 90.97 | 56.02 | 92.30 | 56.53 | 90.56 | 56.59 | 88.50 | 77.53 | 97.18 |
| Internal Conf. | 52.47 | 89.08 | 51.01 | 88.28 | 62.39 | 93.50 | 67.70 | 87.43 | 42.24 | 89.52 |
| SVAR | 54.38 | 90.41 | 65.48 | 89.07 | 67.33 | 94.72 | 65.55 | 85.98 | 55.31 | 87.11 |
| Contextual Lens | 52.49 | 86.43 | 57.98 | 88.15 | 46.95 | 91.37 | 56.59 | 88.50 | 65.21 | 88.94 |
| GLSIM | 70.34 | 93.32 | 59.95 | 90.30 | 68.76 | 95.39 | 65.41 | 88.44 | 70.33 | 90.32 |
| CLS (Ours) | 63.61 | 86.10 | 50.66 | 94.17 | 68.76 | 95.39 | 68.29 | 93.47 | 70.27 | 95.54 |
| CCS (Ours) | 83.57 | 96.62 | 73.41 | 95.90 | 78.23 | 97.28 | 76.37 | 95.72 | 72.64 | 95.25 |
| InsLen (Ours) | 85.58 | 97.74 | 72.75 | 95.77 | 79.82 | 97.48 | 77.22 | 96.63 | 75.27 | 95.60 |

that of the Context Consistency Score, where top-$k$ instruction embeddings are explicitly used to estimate the reliability of instruction selection. In contrast, the Calibration Confidence is intended to measure the model's confidence in the visual evidence of the object. Therefore, the maximum operation better preserves strong visual cues, while averaging over multiple embeddings may introduce noise from less informative instructions.

**Instruction templates in Table 6.** In the experiments reported in Table 6, we evaluate instructions with varying lengths but identical semantic content. The specific instruction templates are provided in Table 14.

### A.3. Implementation details.

We implement all MLLMs with greedy decoding and conduct all experiments on NVIDIA DRIVE-PG199-RPOD GPUs with 32 GB memory each. The Local Similarity Score (LSS) is adopted as the vision-based detector. For hyperparameter settings, the number $K$ of selected image embeddings is set to 32 for LLaVA-1.5, 4 for InstructBLIP and mPLUG-Owl3, and 16 for LLaVA-OneVision1.5 and Qwen3-VL. In particular, when sampler modules (e.g., Q-Former) are used, a smaller $K$ is adopted due to the reduced number of visual tokens. The decoder layer $l'$ from which image embeddings are extracted is set to 32 for LLaVA-1.5, 30 for InstructBLIP, 35 for Qwen3-VL and LLaVA-OneVision1.5, and 27 for mPLUG-Owl3.

### A.4. Case study of interpreting instruction embeddings.

In this section, we extend our qualitative example across all four architectures to demonstrate our finding about instruction embeddings for visual evidence. On the one hand, Figures 14–16 present the interpretation results of instruction embeddings

*Table 12.* Performance comparison on larger models on the MSCOCO benchmark.

| Method | LLaVA-1.5-13B | | InstructBLIP-13B | |
|---|---|---|---|---|
| | AUROC | AUPR | AUROC | AUPR |
| GLSIM | 85.36 | 95.37 | 84.77 | 95.01 |
| Internal Conf. | 73.66 | 91.67 | 72.34 | 89.90 |
| InsLen | **89.62** | **96.27** | **89.04** | **95.46** |

*Table 13.* Performance comparison of different design variants of the Context Consistency Score on the MSCOCO dataset.

| Method | Original | | Cos | | Distance | | Direction | |
|---|---|---|---|---|---|---|---|---|
| | AUROC ↑ | AUPR ↑ | AUROC ↑ | AUPR ↑ | AUROC ↑ | AUPR ↑ | AUROC ↑ | AUPR ↑ |
| LLaVA-1.5 | 82.72 | 93.83 | 81.04 | 93.24 | 58.40 | 79.65 | 72.95 | 90.98 |
| Qwen3-VL | 79.07 | 95.06 | 77.70 | 95.21 | 78.05 | 95.32 | 61.37 | 76.33 |

using the Logit Lens technique for LLaVA-1.5, LLaVA-OneVision1.5, and Qwen3-VL, respectively. For example, in LLaVA-OneVision1.5, the tokens decoded from instruction embeddings contain abundant image-related information (highlighted in blue), such as "Baseball" and "MLB". On the other hand, these visually grounded tokens are associated with higher probabilities of corresponding real objects in the image. Following a similar protocol as in Section 4.1, we further examine the internal confidence derived from instruction embeddings and image embeddings, with results shown in Figure 9. We observe that confidence estimated from instruction embeddings more effectively distinguishes hallucinated samples from grounded ones. These case studies are also consistent with our findings that instruction embeddings can help filter misleading visual information.

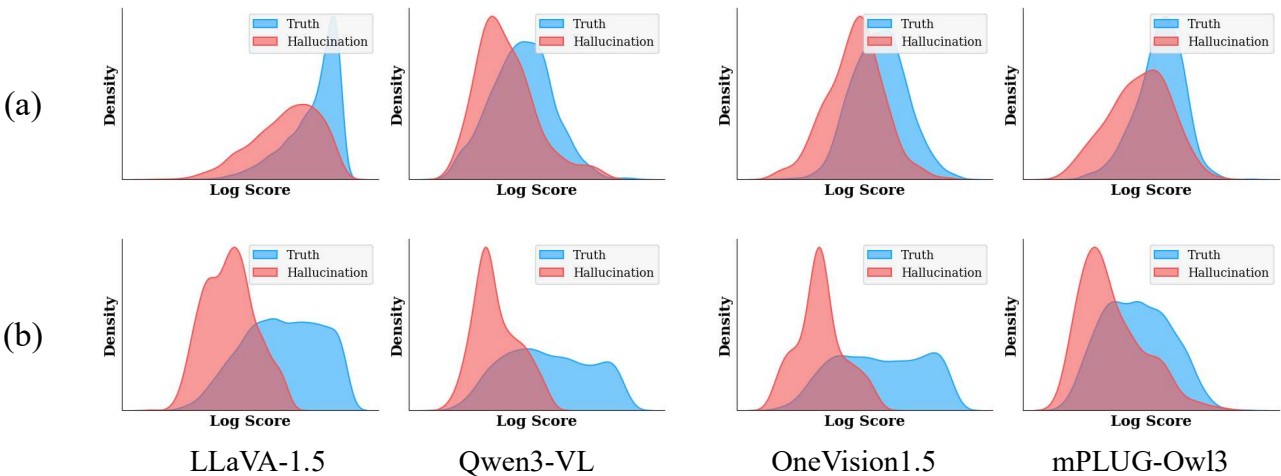

*Figure 9.* Density distributions of log-transformed confidence scores derived from image embeddings (a) and instruction embeddings (b), where log transformation is used for clearer comparison.

## A.5. Case study of object detection.

We present qualitative examples of real-time deployment of our InsLen score for object hallucination detection in Figures 12 and 13. The decision thresholds defined in Section 3 are determined on a validation set of 500 samples, and are set to $9.4 \times 10^{-6}$ for InsLen and $9.3 \times 10^{-2}$ for GLSIM on LLaVA-1.5-7B, and $1.58 \times 10^{-6}$ for InsLen and $1.37 \times 10^{-2}$ for GLSIM on Qwen3-VL, respectively. As shown in Figure 12, InsLen correctly identifies the hallucinated object *spoon*, while GLSIM fails due to locally similar visual patterns. Overall, InsLen can more reliably distinguish hallucinated objects that share similar local appearances by jointly leveraging global object-level contextual cues and calibrated local visual evidence.

*Table 14.* Performance comparison on POPE benchmarks.

| Instruction type | Content |
|---|---|
| Original instruction | "Please describe the image in detail." |
| Short instruction | 1)"Detail the image." 
 2) "Describe all details." 
 3) "Describe the image.". |
| Long instruction | 1) "Please first roughly observe the main objects in the picture, and then carefully examine all the objects. And then describe the given image in detail." 

 2) "First capture the general impression and primary objects of the image. Then analyze every visible detail step by step. Finally, produce a complete and detailed account of the image." 

 3) "Begin with a rough summary of the key elements in the picture. After that, closely inspect every part of the image, including small details. Conclude with a full and detailed description." |

## A.6. Limitation and Further work.

**Limitation.** In certain models, such as InstructBLIP, visual and instruction information are fused prior to being processed by the LLM. Specifically, InstructBLIP employs a Q-Former that is extensively pretrained with large-scale image–text alignment objectives, endowing it with a strong capability to filter erroneous visual features originating from the visual encoder. In this setting, the subsequent processing by the LLM may further suppress informative visual signals, leading to a degradation in detection performance. We conduct probing experiments similar to those in Figure 3 and observe that confidence scores derived from "image embeddings" achieve an AUROC of 80%, outperforming those derived from instruction embeddings (72%).

**Further work.** The properties of instruction embeddings demonstrated in this work show strong potential for object hallucination detection. While our study primarily focuses on the OH detection task, these properties could be further explored in broader directions, such as guiding decoding strategies and informing post-training techniques aimed at mitigating hallucinations in MLLMs.

## B. Baselines.

**Negative Log-likelihood (NLL).** NLL (Zhou et al., 2024) is proposed to assess hallucination by exploiting the confidence of object token generation in an autoregressive manner. Intuitively, if an object is weakly supported by visual evidence, the model assigns it a low decoding probability. Formally, consider an object token $\mathbf{o}$ produced at decoding step $j$, its conditional probability is given by $p(\mathbf{o} \mid \mathbf{y}_{<j}, \mathbf{v})$. To align with the definition in Eq. (1), the hallucination score for object $\mathbf{o}$ is then defined as:

$$s_{\text{nll}} = \log p(\mathbf{o} \mid \mathbf{y}_{<j}, \mathbf{v}). \tag{9}$$

**Entropy.** Entropy (Malinin & Gales, 2021) is proposed to characterize object-level hallucination by measuring the overall uncertainty of the token distribution produced during autoregressive decoding. Instead of focusing on a single predicted token, this metric considers the entire vocabulary-level distribution at decoding step $j$. Concretely, to align with the definition in Eq. (1), the hallucination score is computed as:

$$s_{\text{entropy}} = \sum_{\mathbf{y} \in \mathcal{V}} p(\mathbf{y} \mid \mathbf{y}_{<j}, \mathbf{v}) \log p(\mathbf{y} \mid \mathbf{y}_{<j}, \mathbf{v}). \tag{10}$$

**Internal Confidence.** Internal Confidence (Jiang et al., 2025a) is proposed to assess object-level hallucination by probing the confidence encoded in intermediate image representations via the logit lens. Instead of relying on final decoding outputs, this approach projects visual features at different layers into the language vocabulary space and examines their alignment

with object words. Concretely, the internal confidence for an object token $o$ is obtained by taking the maximum softmax probability assigned to $o$ across all image patches and decoder layers. The hallucination score is defined as:

$$s_{\text{IC}} = \max_{l=1,\ldots,L} \max_{i=1,\ldots,N} \text{VLL}_l(\mathbf{v}_i)[\mathbf{o}], \tag{11}$$

where $L$ refers to the number of decoder layers and $N$ refers to the number of image patches.

**Summed Visual Attention Ratio (SVAR).** SVAR ([Jiang et al., 2025b](#)) is proposed to quantify object-level hallucination by examining how strongly a generated object token attends to visual tokens across decoder layers. The core intuition is that tokens grounded in visual evidence tend to exhibit higher attention weights toward image patches. To this end, the Visual Attention Ratio (VAR) is first defined to measure the amount of attention an object token allocates to image tokens within a specific attention head and layer. Based on VAR, the Summed Visual Attention Ratio (SVAR) aggregates visual attention signals by averaging over all heads and accumulating contributions across a designated range of middle layers.

Concretely, for an object token $o$, SVAR is computed by summing VAR scores from layers $l_5$ to $l_{18}$ and normalizing over attention heads:

$$s_{\text{SVAR}} = \frac{1}{H} \sum_{l=5}^{18} \sum_{h=1}^{H} \text{VAR}^{(l,h)}(\mathbf{o}), \tag{12}$$

$$\text{VAR}^{(l,h)}(\mathbf{o}) \triangleq \sum_{i=1}^{N} A^{(l,h)}(\mathbf{o}, \mathbf{o}_i), \tag{13}$$

where $H$ denotes the total number of attention heads, and $A^{(l,h)}(\mathbf{o}, \mathbf{v}_i)$ represents the attention weight from object token $\mathbf{o}$ to image token $\mathbf{v}_i$ at $h$-th head in $l$-th layer.

**Contextual Lens.** Contextual Lens ([Phukan et al., 2025](#)) is proposed to measure hallucination by evaluating the semantic alignment between generated tokens and visual representations using contextual embeddings from intermediate layers. The key idea is to assess whether a generated token is supported by any image patch in the shared embedding space. At the sentence level, the method computes the similarity between the averaged embedding of all generated tokens at a designated textual layer and each image patch embedding extracted from a specified visual layer, taking the maximum similarity as the confidence score.

To adapt this formulation for object-level hallucination detection, we replace the sentence embedding with the embedding of a single object token. Specifically, the hallucination score is defined as:

$$s_{\text{CL}} = \max_{i=1,\ldots,N} \text{sim}\big(\mathbf{h}_{l_T}(\mathbf{o}), \mathbf{h}_{l_I}(\mathbf{o}_i)\big), \tag{14}$$

where $\mathbf{h}_{l_T}(\mathbf{o})$ denotes the contextual embedding of the object token at layer $l_T$, and $\mathbf{h}_{l_I}(\mathbf{v}_i)$ represents the embedding of the $i$-th image patch at layer $l_I$.

**EAZY.** EAZY ([Che et al., 2025](#)) proposes a training-free approach to object hallucination detection by identifying hallucinatory image tokens that spuriously trigger object generation. The method is motivated by the observation that hallucinated objects are often caused by a small number of visually biased image tokens, which receive abnormally high attention from the decoder during generation, despite lacking proper semantic grounding.

Specifically, EAZY analyzes the cross-attention patterns between generated object tokens and image tokens at intermediate decoder layers. Image tokens that consistently attract disproportionately large attention weights are identified as hallucinatory image tokens. To verify whether such tokens are responsible for hallucination, EAZY performs targeted interventions by suppressing these tokens during decoding and re-generating the response. If the object disappears after the intervention, it is deemed hallucinated; otherwise, it is considered visually grounded.

## C. Benchmark details.

**CLEVR.** CLEVR (Johnson et al., 2017) is a synthetic VQA benchmark for diagnostic evaluation of compositional visual reasoning. The benchmark contains approximately 1,000,000 questions, which can be grouped into several major question types, including Query Attribute (*e.g.*, color, shape, size, and material), Compare Attribute, Existence, Counting, and Integer Comparison (*e.g.*, less than, greater than, and equal). In our experiments, we randomly sample 5,000 questions from CLEVR, covering all major question types. We prompt the LVLM with the template as shown in Figure 10.

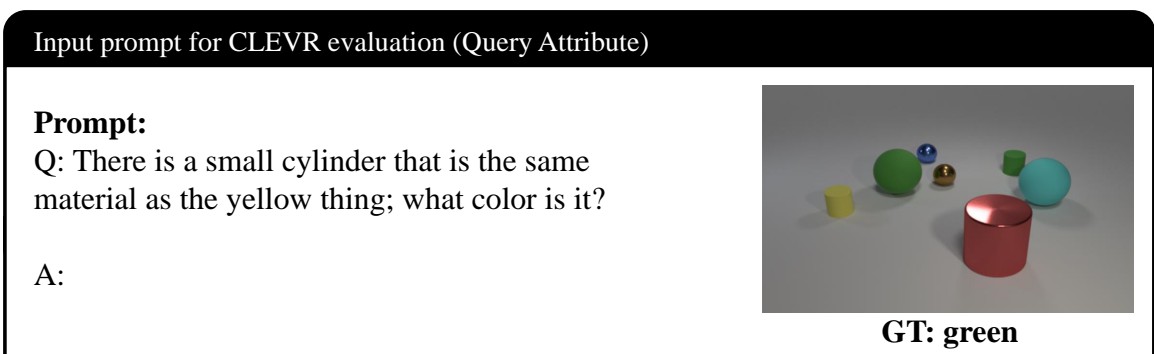

*Figure 10.* Input prompt for CLEVR evaluation (Query Attribute).

**POPE.** POPE (Li et al., 2023) proposes a benchmark for evaluating object hallucination in LVLM. It formulates hallucination evaluation as a binary probing task, where models are prompted with Yes/No questions about the presence of specific objects in an image. The benchmark further supports multiple object sampling strategies—Random, Popular, and Adversarial—to assess models' tendencies to hallucinate frequently appearing or co-occurring objects. Responses containing "Yes" are labeled as 1 (real), and all others as 0 (hallucinated). For evaluation, each question is labeled using the ground-truth object annotations from the MSCOCO dataset. We prompt the LVLM with the template as shown in Figure 11.

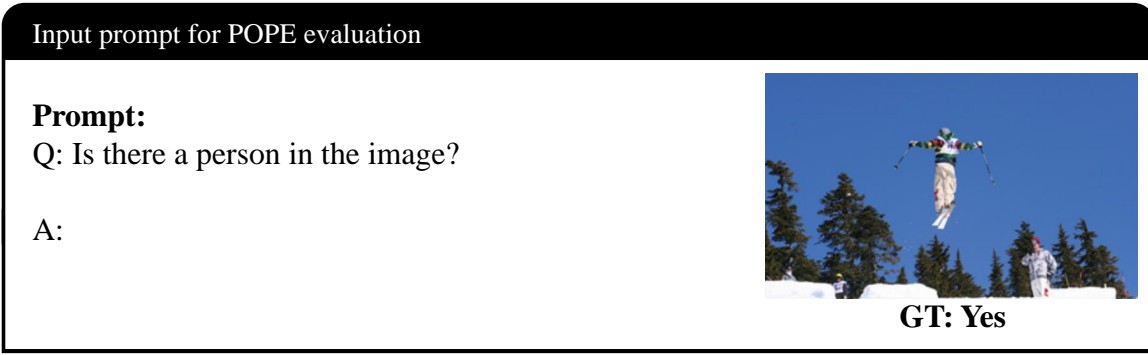

*Figure 11.* Input prompt for POPE evaluation.

**[Input image]** :

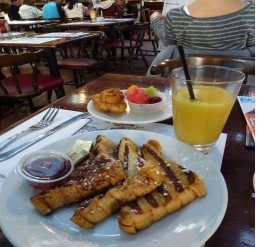

User :
**Please describe the image in detail.**

🌋 LLaVA-1.5 :

**The image features a dining table with a plate of food and a glass of orange juice. The plate contains a variety of food items, including a sandwich, a pastry, and a bowl of fruit. The sandwich is placed towards the center of the plate, while the pastry is situated on the left side. The bowl of fruit is located on the right side of the plate.**
**In addition to the main dish, there are several utensils on the table, including a fork on the left side, a knife on the right side, and a spoon on the far left. A cup can also be seen on the right side of the table. The table is surrounded by chairs, with one on the left side and another on the right side.**

InsLen (Ours)

**The image features a dining table with a plate of food and a glass of orange juice. The plate contains a variety of food items, including a sandwich, a pastry, and a bowl of fruit. The sandwich is placed towards the center of the plate, while the pastry is situated on the left side. The bowl of fruit is located on the right side of the plate.**
**In addition to the main dish, there are several utensils on the table, including a fork on the left side, a knife on the right side, and a spoon on the far left. A cup can also be seen on the right side of the table. The table is surrounded by chairs, with one on the left side and another on the right side.**

GLSIM

**The image features a dining table with a plate of food and a glass of orange juice. The plate contains a variety of food items, including a sandwich, a pastry, and a bowl of fruit. The sandwich is placed towards the center of the plate, while the pastry is situated on the left side. The bowl of fruit is located on the right side of the plate.**
**In addition to the main dish, there are several utensils on the table, including a fork on the left side, a knife on the right side, and a spoon on the far left. A cup can also be seen on the right side of the table. The table is surrounded by chairs, with one on the left side and another on the right side.**

**[Input image]** :

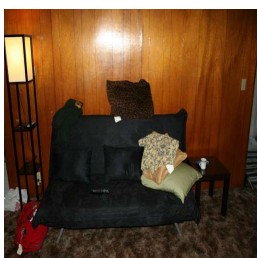

User :
**Please describe the image in detail.**

🌋 LLaVA-1.5 :

**The image features a black couch with a remote control placed on it. The couch is situated in a living room, and there are several pillows on it, providing a cozy and inviting atmosphere.**
**In addition to the couch, there is a dining table located in the room, with a cup and a spoon placed on it. A chair can be seen near the table, and a book is also present in the scene, possibly indicating a relaxing reading session.**

InsLen (Ours)

**The image features a black couch with a remote control placed on it. The couch is situated in a living room, and there are several pillows on it, providing a cozy and inviting atmosphere.**
**In addition to the couch, there is a dining table located in the room, with a cup and a spoon placed on it. A chair can be seen near the table, and a book is also present in the scene, possibly indicating a relaxing reading session.**

GLSIM

**The image features a black couch with a remote control placed on it. The couch is situated in a living room, and there are several pillows on it, providing a cozy and inviting atmosphere.**
**In addition to the couch, there is a dining table located in the room, with a cup and a spoon placed on it. A chair can be seen near the table, and a book is also present in the scene, possibly indicating a relaxing reading session.**

*Figure 12.* Qualitative comparison between InsLen and GLSIM on object hallucination detection where the LLaVA-1.5-7B model is used. In the generated responses (right), ground-truth objects are shown in green and hallucinated objects in orange. Detection results are indicated by green (real) and orange (hallucinated), and incorrect predictions are marked with a ✗.

**[Input image]** :

**User** :
Please describe the image in detail.

**Qwen3-VL** :

This is a close-up, appetizing photograph of a freshly baked, round **pizza** resting on a dark, round wooden serving board. The **pizza** has a thick, rustic crust that is golden-brown and slightly puffy, with some charred spots suggesting it was baked in a hot **oven**.
…
In the background, slightly out of focus, are elements of a dining setting: a blue **plate** on the left, a white **napkin**, and a wooden knife block or utensil holder. The overall lighting is warm and natural, enhancing the cozy, comforting appeal of the meal. The image captures the **pizza** from a slightly elevated angle, making it the clear focal point.

InsLen (Ours)

This is a close-up, appetizing photograph of a freshly baked, round **pizza** resting on a dark, round wooden serving board. The **pizza** has a thick, rustic crust that is golden-brown and slightly puffy, with some charred spots suggesting it was baked in a hot **oven**.
…
In the background, slightly out of focus, are elements of a dining setting: a blue **plate** on the left, a white **napkin**, and a wooden knife block or utensil holder. The overall lighting is warm and natural, enhancing the cozy, comforting appeal of the meal. The image captures the **pizza** from a slightly elevated angle, making it the clear focal point.
right side.

GLSIM

This is a close-up, appetizing photograph of a freshly baked, round **pizza** resting on a dark, round wooden serving board. The **pizza** has a thick, rustic crust that is golden-brown and slightly puffy, with some charred spots suggesting it was baked in a hot **oven**.
…
In the background, slightly out of focus, are elements of a dining setting: a blue **plate** on the left, a white **napkin**, and a wooden knife block or utensil holder. The overall lighting is warm and natural, enhancing the cozy, comforting appeal of the meal. The image captures the **pizza** from a slightly elevated angle, making it the clear focal point.
right side.

**[Input image]** :

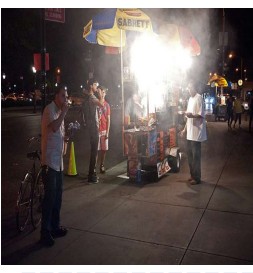

**User** :
Please describe the image in detail.

**Qwen3-VL** :
The main focus is a mobile food **cart**, likely selling **hot dogs** or sausages, branded with the name "SABRETT" in large, bold letters on its canopy. The canopy itself is a colorful **umbrella** with blue, yellow, and red sections. …Several **people** are gathered around the **cart**: On the left, a man in a light-colored, short-sleeved shirt and dark pants is standing with a **bicycle** next to him. He is holding a drink or food item to his mouth, appearing to eat or drink. In the center, a man in a grey hoodie and dark pants is holding a **phone** up, seemingly taking a photo or video of the **cart** or the food

InsLen (Ours)

The main focus is a mobile food **cart**, likely selling **hot dogs** or sausages, branded with the name "SABRETT" in large, bold letters on its canopy. The canopy itself is a colorful **umbrella** with blue, yellow, and red sections. …Several **people** are gathered around the **cart**: On the left, a man in a light-colored, short-sleeved shirt and dark pants is standing with a **bicycle** next to him. He is holding a drink or food item to his mouth, appearing to eat or drink. In the center, a man in a grey hoodie and dark pants is holding a **phone** up, seemingly taking a photo or video of the **cart** or the food

GLSIM

The main focus is a mobile food **cart**, likely selling **hot dogs** or sausages, branded with the name "SABRETT" in large, bold letters on its canopy. The canopy itself is a colorful **umbrella** with blue, yellow, and red sections. …Several **people** are gathered around the **cart**: On the left, a man in a light-colored, short-sleeved shirt and dark pants is standing with a **bicycle** next to him. He is holding a drink or food item to his mouth, appearing to eat or drink. In the center, a man in a grey hoodie and dark pants is holding a **phone** up, seemingly taking a photo or video of the **cart** or the food

*Figure 13.* Qualitative comparison between InsLen and GLSIM on object hallucination detection where the Qwen3-VL model is used. In the generated responses (right), ground-truth objects are shown in green and hallucinated objects in orange. Detection results are indicated by green (real) and orange (hallucinated), and incorrect predictions are marked with a ✗.

[Input image]:

User:

Please describe the image in detail.

🌋 LLaVA-OneVision-1.5:

he image depicts a baseball game in progress, focusing on two key players: the batter and the catcher. The scene is set on a well-maintained baseball field with green grass and dirt around the home plate area. **Batter:** - **Uniform:** The batter is wearing a white uniform with red accents. The number "13" is prominently displayed on his back. - **Helmet:** He has a gold-colored helmet on for protection. - **Glove:** His left hand is equipped with a black batting glove. - **Bat:** The batter …

🔲 **<Please>**

✂ [' describe', '描述', '\tdescribe', 'describe', 'Describe', ' descriptive', '.describe', '描写', ' Describe', ' descr', ' descri', ' description', ' describing', ' describes', ' setDescription', ' caption', ' descriptions', ' besch', 'Descriptions', ' write']

🔲 **< describe>**

✂ [' image', ' 图片', ' detailed', ' images', ' Image', '这张', 'image', '-image', '详细', ' Images', '图片来源', ' imagem', '\timage', ' getImage', '_image', ' imagen', '圖片', 'Image', '图像', '\tImage'] ('职业')

🔲 **< the>**

✂ [' image', ' scene', ' 图片', ' actions', ' images', ' detailed', ' activity', '圖片', ' contents', ' situation', ' relationship', ' entirety', ' location', ' picture', ' activities', '图片来源', ' entire', ' Images', 'scene', ' activities'] ('白衣', ' Baseball', ' authenticity')

🔲 **< image>**

✂ [' detailed', '详细', ' 内容', ' you', ' depicted', '的内容', ' подроб', ' content', '您', ' 具体内容', ' contents', ' portrayed', '内容', '细致', 'あなた', '你看', 'Detailed', 'etailed', '你也', '详细的']

🔲 **< in>**

✂ [' detail', '详细', 'detail', '-detail', ' Detail', ' details', '.detail', ' detal', ' detailed', 'Detail', '_detail', '细节', '(detail', ' Details', '详', '詳細', ' detailing', '/detail', ' détail', 'Details']

🔲 **< detail>**

✂ [' detailing', ' detail', '(detail', '细节', 'detail', ' based', ' including', ' INCLUDING', '.detail', '.Detail', ' Detail', '/detail', ' Including', ' détail', '包括', ' detailed', ' Based', ' focusing', '详细', ' incorporating']

🔲 **<.>**

✂ [' DON', ' Avoid', ' Include', '记住', 'Avoid', 'DON', ' focus', ' Focus', 'Include', '聚焦', ' Including', '记得', '不要', ' focus', ' Don', '避免', ' remember', '.focus', ' OCR', ' include']

🔲 **<|im_end|>**

✂ [' baseball', ' Baseball', ' softball', ' 棒', '🚫', ' MLB', '正在进行', ' soccer', ' boys', '男孩子', ' 少年', ' youth', ' volleyball', ' pitchers', ' fastball', '青少年', '小学生', ' pitcher', '球员', ' gameplay']

🔲 **<\n>**

✂ ['描述', ' Certainly', '这张', ' Certainly', ' apologies', ' apologize', '这段', '抱歉', '', ' Description', ' description', ' describe', ' Imag', '的照片', '_caption', '图片', '这场比赛', '-description', 'This', ' capturing']

🔲 **<|im_start|>**

✂ ['球员', '号码', ' baseball', '编号', ' номер', 'प्रार्थ करह', ' Baseball', ' player', ' number', '号', ' softball', ' jersey', ' players', ' playerId', ' soccer', '球队', ' número', ' Player', '.number', ' Players'] ('手套', ' Shoe', ' sneakers')

🔲 **<assistant>**

✂ ['描述', ' describe', '-description', '_Image', ' Imag', 'DESCRIPTION', ' perceive', ' edes', ':image', '我就', ' 图片', ' Perception', '<Image', ' Description', 'Markdown', '_image', 'Imagine', ' description', '这段', 'markdown'] ('两名', '的强大', ' número')

🔲 **<\n>**

✂ ['The', ' baseball', '🚫', '两名', 'This', ' Baseball', '这场比赛', ' Picture', ' The', '两位', 'Certainly', '球员', '两个人', ' ump', 'Two', '球场', '_two', '(images', ':The', '該'] (' sport', '玩家们', '_players')

⚙ Logit lens result    🔲 Instruction embeddings

*Figure 14.* Visualization of the instruction embeddings in LLaVA-Oneversion-1.5, where the input instruction (*e.g.*, "Please describe the image in detail.") is projected back to the vocabulary space. For each instruction token position, we report the top-20 vocabulary tokens with the highest prediction scores, revealing the semantic distribution induced by the instruction embeddings. Notably, several instruction-related tokens (highlighted in blue) show strong alignment with visual concepts present in the input image, indicating that the instruction embeddings capture image-related semantic information. (Tokens in parentheses correspond to the top-100 results.)

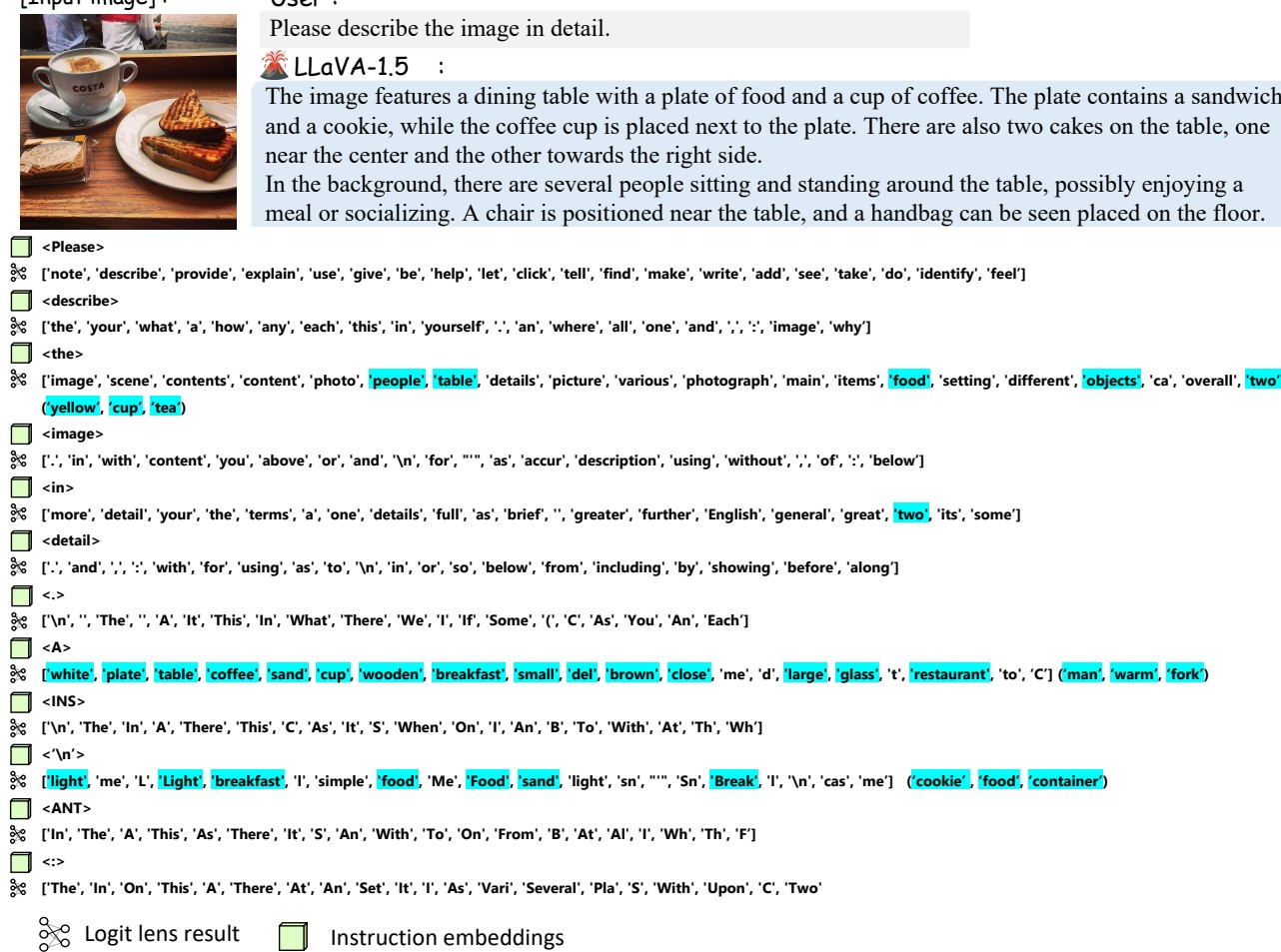

[Input image] :

User :

Please describe the image in detail.

🌋 LLaVA-1.5 :

The image features a dining table with a plate of food and a cup of coffee. The plate contains a sandwich and a cookie, while the coffee cup is placed next to the plate. There are also two cakes on the table, one near the center and the other towards the right side.
In the background, there are several people sitting and standing around the table, possibly enjoying a meal or socializing. A chair is positioned near the table, and a handbag can be seen placed on the floor.

◻ **<Please>**
✄ ['note', 'describe', 'provide', 'explain', 'use', 'give', 'be', 'help', 'let', 'click', 'tell', 'find', 'make', 'write', 'add', 'see', 'take', 'do', 'identify', 'feel']

◻ **<describe>**
✄ ['the', 'your', 'what', 'a', 'how', 'any', 'each', 'this', 'in', 'yourself', '.', 'an', 'where', 'all', 'one', 'and', ',', ':', 'image', 'why']

◻ **<the>**
✄ ['image', 'scene', 'contents', 'content', 'photo', 'people', 'table', 'details', 'picture', 'various', 'photograph', 'main', 'items', 'food', 'setting', 'different', 'objects', 'ca', 'overall', 'two'] ('yellow', 'cup', 'tea')

◻ **<image>**
✄ ['.', 'in', 'with', 'content', 'you', 'above', 'or', 'and', '\n', 'for', '"'", 'as', 'accur', 'description', 'using', 'without', ',', 'of', ':', 'below']

◻ **<in>**
✄ ['more', 'detail', 'your', 'the', 'terms', 'a', 'one', 'details', 'full', 'as', 'brief', '', 'greater', 'further', 'English', 'general', 'great', 'two', 'its', 'some']

◻ **<detail>**
✄ ['.', 'and', ',', ':', 'with', 'for', 'using', 'as', 'to', '\n', 'in', 'or', 'so', 'below', 'from', 'including', 'by', 'showing', 'before', 'along']

◻ **<.>**
✄ ['\n', '', 'The', '', 'A', 'It', 'This', 'In', 'What', 'There', 'We', 'I', 'If', 'Some', '(', 'C', 'As', 'You', 'An', 'Each']

◻ **<A>**
✄ ['white', 'plate', 'table', 'coffee', 'sand', 'cup', 'wooden', 'breakfast', 'small', 'del', 'brown', 'close', 'me', 'd', 'large', 'glass', 't', 'restaurant', 'to', 'C'] ('man', 'warm', 'fork')

◻ ****
✄ ['\n', 'The', 'In', 'A', 'There', 'This', 'C', 'As', 'It', 'S', 'When', 'On', 'I', 'An', 'B', 'To', 'With', 'At', 'Th', 'Wh']

◻ **<'\n'>**
✄ ['light', 'me', 'L', 'Light', 'breakfast', 'I', 'simple', 'food', 'Me', 'Food', 'sand', 'light', 'sn', '"'", 'Sn', 'Break', 'I', '\n', 'cas', 'me'] ('cookie', 'food', 'container')

◻ **<ANT>**
✄ ['In', 'The', 'A', 'This', 'As', 'There', 'It', 'S', 'An', 'With', 'To', 'On', 'From', 'B', 'At', 'Al', 'I', 'Wh', 'Th', 'F']

◻ **<:>**
✄ ['The', 'In', 'On', 'This', 'A', 'There', 'At', 'An', 'Set', 'It', 'I', 'As', 'Vari', 'Several', 'Pla', 'S', 'With', 'Upon', 'C', 'Two']

✄ Logit lens result ◻ Instruction embeddings

*Figure 15.* Visualization of the instruction embeddings in LLaVA-1.5.

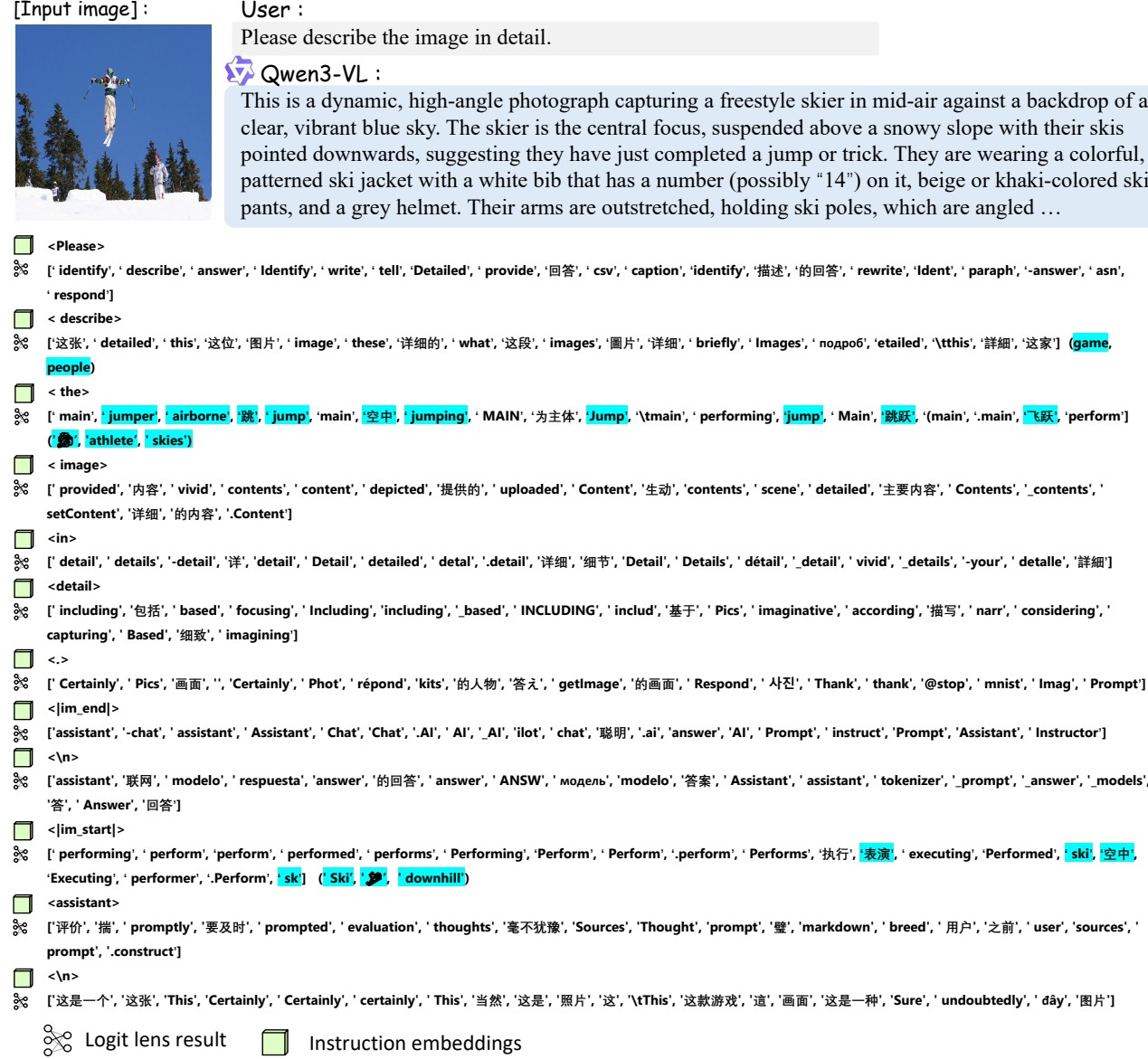

**[Input image]:**

**User:**
Please describe the image in detail.

**Qwen3-VL:**
This is a dynamic, high-angle photograph capturing a freestyle skier in mid-air against a backdrop of a clear, vibrant blue sky. The skier is the central focus, suspended above a snowy slope with their skis pointed downwards, suggesting they have just completed a jump or trick. They are wearing a colorful, patterned ski jacket with a white bib that has a number (possibly "14") on it, beige or khaki-colored ski pants, and a grey helmet. Their arms are outstretched, holding ski poles, which are angled …

**<Please>**
['identify', 'describe', 'answer', 'Identify', 'write', 'tell', 'Detailed', 'provide', '回答', 'csv', 'caption', 'identify', '描述', '的回答', 'rewrite', 'Ident', 'paraph', '-answer', 'asn', 'respond']

**< describe>**
['这张', 'detailed', 'this', '这位', '图片', 'image', 'these', '详细的', 'what', '这段', 'images', '圖片', '详细', 'briefly', 'Images', 'подроб', 'etailed', '\tthis', '詳細', '这家'] (game, people)

**< the>**
['main', 'jumper', 'airborne', '跳', 'jump', 'main', '空中', 'jumping', 'MAIN', '为主体', 'Jump', '\tmain', 'performing', 'jump', 'Main', '跳跃', '(main', '.main', '飞跃', 'perform'] (' 🏂', 'athlete', 'skies')

**< image>**
['provided', '内容', 'vivid', 'contents', 'content', 'depicted', '提供的', 'uploaded', 'Content', '生动', 'contents', 'scene', 'detailed', '主要内容', 'Contents', '_contents', 'setContent', '详细', '的内容', '.Content']

**<in>**
['detail', 'details', '-detail', '详', 'detail', 'Detail', 'detailed', 'detal', '.detail', '详细', '细节', 'Detail', 'Details', 'détail', '_detail', 'vivid', '_details', '-your', 'detalle', '詳細']

**<detail>**
['including', '包括', 'based', 'focusing', 'Including', 'including', '_based', 'INCLUDING', 'includ', '基于', 'Pics', 'imaginative', 'according', '描写', 'narr', 'considering', 'capturing', 'Based', '细致', 'imagining']

**<.>**
['Certainly', 'Pics', '画面', '', 'Certainly', 'Phot', 'répond', 'kits', '的人物', '答え', 'getImage', '的画面', 'Respond', '사진', 'Thank', 'thank', '@stop', 'mnist', 'Imag', 'Prompt']

**<|im_end|>**
['assistant', '-chat', 'assistant', 'Assistant', 'Chat', 'Chat', '.AI', 'AI', '_AI', 'ilot', 'chat', '聪明', '.ai', 'answer', 'AI', 'Prompt', 'instruct', 'Prompt', 'Assistant', 'Instructor']

**<\n>**
['assistant', '联网', 'modelo', 'respuesta', 'answer', '的回答', 'answer', 'ANSW', 'модель', 'modelo', '答案', 'Assistant', 'assistant', 'tokenizer', '_prompt', '_answer', '_models', '答', 'Answer', '回答']

**<|im_start|>**
['performing', 'perform', 'perform', 'performed', 'performs', 'Performing', 'Perform', 'Perform', '.perform', 'Performs', '执行', '表演', 'executing', 'Performed', 'ski', '空中', 'Executing', 'performer', '.Perform', 'sk'] ('Ski', '🏂', 'downhill')

**<assistant>**
['评价', '搞', 'promptly', '要及时', 'prompted', 'evaluation', 'thoughts', '毫不犹豫', 'Sources', 'Thought', 'prompt', '壁', 'markdown', 'breed', '用户', '之前', 'user', 'sources', 'prompt', '.construct']

**<\n>**
['这是一个', '这张', 'This', 'Certainly', 'Certainly', 'certainly', 'This', '当然', '这是', '照片', '这', '\tThis', '这款游戏', '這', '画面', '这是一种', 'Sure', 'undoubtedly', 'đây', '图片']

Logit lens result     Instruction embeddings

*Figure 16.* Visualization of the instruction embeddings in Qwen3-VL.

