# OpenReview forum: "Instruction Lens Score: Your Instruction Contributes a Powerful Object Hallucination Detector for Multimodal Large Language Models"
_ICML.cc/2026/Conference — ICML 2026 regular_

### Official Review · Reviewer_V8kB · 2026-03-09

**Soundness:** 2
**Presentation:** 2
**Significance:** 2
**Originality:** 2
**Overall Recommendation:** 3
**Confidence:** 4

**Summary:**

This paper proposes a new method, called Instruction Lens Score (InsLen), to detect object hallucinations in multimodal large language models (MLLMs). The authors observe that instruction embeddings may implicitly filter misleading visual information introduced by image embeddings and leverage this property to design a hallucination detection mechanism. Specifically, the proposed InsLen score combines a Calibrated Local Score and a Context Consistency Score to jointly utilize local visual evidence and global contextual information. Through experiments on multiple benchmarks, including MSCOCO, Objects365, POPE, and CLEVR, and across several MLLM architectures, the authors demonstrate that the proposed method achieves improved performance compared to existing hallucination detection approaches.

**Compliance With Llm Reviewing Policy:**

Affirmed.

**Final Justification:**

Although I think the method proposed by the authors is somewhat verbose in the submitted version, they made an effort during the rebuttal process by presenting ideas to improve these aspects. Accordingly, I have adjusted my score.

**Key Questions For Authors:**

Please refer to the Weaknesses section for detailed concerns.

In particular, beyond the empirical observations presented in the paper, it seems crucial to provide a clearer theoretical justification or background explaining why instruction embeddings can effectively filter misleading visual information or be related to the cause of object hallucination. Strengthening this aspect would help clarify the core assumption and improve the overall persuasiveness of the paper.

In addition, the proposed method consists of several heuristic components and multiple processing steps. It may be worth considering whether a simpler and more principled approach with fewer hyperparameters could achieve similar or better performance. If possible, the authors are encouraged to discuss potential directions for simplifying and refining the method to make it more concise and robust.

**Limitations:**

No. See the Weaknesses section and Key Questions for Authors section.

**Strengths And Weaknesses:**

**Strengths**

1.The paper presents an interesting perspective by analyzing the role of instruction embeddings in multimodal large language models and exploring how they can be utilized for hallucination detection. In particular, the observation that instruction embeddings may suppress misleading signals originating from visual embeddings provides a useful insight into the internal behavior of MLLMs. This analysis could potentially inspire further research on understanding and interpreting multimodal representations.

2.The proposed method is training-free and plug-and-play, meaning that it can be applied without additional model training or external auxiliary models. Compared to prior approaches that rely on large external models such as GPT-4 for hallucination detection, this design makes the method computationally lighter and easier to integrate into existing MLLM systems.

3.The paper also presents relatively comprehensive experimental evaluations across multiple models and benchmarks. Experiments are conducted on several MLLM architectures, including LLaVA, InstructBLIP, Qwen-VL, and mPLUG-Owl, as well as multiple evaluation datasets such as MSCOCO, Objects365, POPE, and CLEVR. In addition, the authors include ablation studies and sensitivity analyses that provide further insights into the contributions of the individual components of the proposed method.

**Weaknesses**

1.The primary concern lies in the core assumption that instruction embeddings can effectively filter misleading visual information. Although this claim is empirically supported throughout the paper, it does not appear to be grounded in a clear theoretical justification. For instance, a generic instruction such as “Please describe the image in detail.” may reflect language priors more strongly than actual visual evidence. As a result, it is difficult to determine whether the observed performance improvement truly arises from improved visual grounding or from prompt-induced linguistic bias. To better support the main claim, stronger theoretical justification in addition to empirical evidence would be necessary.

2.Another concern is the reliance on heuristically designed scoring components. The InsLen score combines several heuristic elements, including calibration confidence, context consistency score, and the weighting factor 𝜔. Similar to the first issue, many aspects of this design appear to be empirically motivated without clear theoretical grounding. Moreover, since the method depends on multiple hyperparameters, the performance may vary depending on their settings. Although some sensitivity analyses are provided, it remains unclear whether the approach can generalize robustly across different settings without careful tuning.

3.In addition, recent studies (e.g., “Temporal Reasoning Transfer from Text to Video,” ICLR 2025) suggest that hallucination or reasoning issues in MLLMs may originate from the language model itself rather than from image embeddings. Therefore, approaches that attempt to improve MLLM performance by leveraging instruction or text embeddings are not entirely new. It would be beneficial for the paper to more clearly articulate how the proposed method overcomes the limitations of prior work and what aspects of the approach are truly novel compared to existing research.

---

> ### Author Rebuttal · Authors · 2026-03-31
>
> We thank the reviewer for the insightful comments and will address the questions below.
> ## Q1 & W1: Justification for our motivation.
> Regarding the concern that “instructions may primarily reflect language priors”, recent studies [c5,c6] suggest that instructions aggregate both visual and textual cues rather than merely language priors.
> We further conduct Attention Knockout experiments [c7], suppressing instruction–image interactions.
> The result shows that InsLen drops from 86.9% to 75.69% AUROC on MSCOCO benchmark, confirming they encode visual information beyond language priors.
>
> For theoretical justification about “filtering misleading visual information”, previous work [c6] suggests that instruction embeddings play a crucial role in information flow to express visual information.
> From the perspective of the information bottleneck principle [c8], this process can be formulated as an information bottleneck objective: $\max I(H_i, H_a) - \beta I(H_i, H_v)$,
> where $H_v, H_i, H_a$ denote the embeddings of input image, instruction, and answer, respectively; and $I(\cdot,\cdot)$ denotes mutual information.
> During training, $H_i$ serves as a bridge between noisy image embeddings and the correct answer, and learns to filter the misleading visual features.
> ## Q2 & W2 : Justification of our design.
> We clarify that the components in InsLen are not heuristics, but are designed to capture complementary signals for detection.
> As discussed in Introduction Section, hallucinations can arise from misleading visual signals.
> Calibration Confidence corrects unreliable visual evidence.
> Context Consistency Score captures context signals via instruction embeddings without auxiliary models.
> Since these two signals are derived from different evidence (visual reliability and contextual alignment), the weighting factor $\omega$ balances these complementary cues.
>
> We demonstrate the robustness of InsLen across multiple dimensions.
> Specifically, we evaluate our method on different model architectures and apply the same hyperparameter setting across various benchmarks, consistently outperforming baselines.
> Furthermore, our sensitivity analyses (Figure 5,7,8) show that InsLen remains stable across a wide range of hyperparameter values, indicating that it does not rely on careful tuning.
> When adapting to a new architecture, similar to other hallucination detection methods, some hyperparameter adjustment may be needed. In practice, hyperparameters can be obtained by performing light tuning on a small validation set (e.g., 500 samples), which is sufficient to obtain strong performance.
>
> We agree that simplifying the method is important.
> Our results suggest that different components of InsLen can be selectively applied depending on the task: For object hallucination detection, Calibration Confidence can be combined with vision-based scores to provide reliable detection.
> For semantic hallucinations (e.g., relational or attribution errors), Context Consistency Score is more suitable, as shown in Appendix Table 9.
> Importantly, each individual component achieves comparable or better performance than baselines (Appendix Table 8,9,10), indicating that InsLen can be simplified for task-specific purposes without sacrificing effectiveness.
> ## W3 : Language prior & novelty.
> Language prior arises in the process of answer prediction.
> In contrast, instructions are explicitly provided by users, allowing the model to concentrate on understanding the task and encoding task-relevant features into the instruction embeddings.
> From this perspective, the ability of instruction embeddings to capture reliable visual information offers a novel viewpoint for hallucination detection.
>
> Prior methods suffer from two limitations:
> (1)They rely on an auxiliary assistant model for detection, leading to computational overhead;
> and (2)Post-hoc detectors rely on visual features, where errors in these features can lead to incorrect detection.
> The novelty of our work lies in three key aspects to address the limitations:
> (1) New insight: We reveal that instruction embeddings encode visual semantics and can filter misleading visual information.
> (2) New methodology: The proposed InsLen score systematically leverages instruction embeddings to correct misleading vision-based confidence by Calibrated Local Score, and to capture context information by Context Consistency Score.
> (3) Practical advantage: Our InsLen does not rely on external models, and can be directly applied to different MLLMs.
> Therefore, InsLen offers clear novelties compared with prior methods.
>
> [c5] Causal Tracing of Object Representations in Large Vision Language Models: Mechanistic Interpretability and Hallucination Mitigation. In: AAAI, 2026.
>
> [c6] Cross-modal Information Flow in Multimodal Large Language Models. In: CVPR, 2025.
>
> [c7] Dissecting Recall of Factual Associations in Auto-Regressive Language Models. In: EMNLP, 2023.
>
> [c8] Deep Learning and the Information Bottleneck Principle. In: IEEE ITW, 2015.

---

> > ### Author Rebuttal · Reviewer_V8kB · 2026-04-03
> >
> > I have reviewed the authors’ rebuttal and appreciate the effort. However, to be candid, the response felt more like a repetition of the original claims rather than a discussion addressing the theoretical justifications I raised or possible ways to simplify the method.
> >
> > My concern is not so much about doubting the necessity or effectiveness of each component, but rather that the current structure appears somewhat complex and relies on multiple heuristic elements. I do acknowledge that the usefulness of each component has been demonstrated experimentally, as presented.
> >
> > At this stage, I would like to maintain my original score.

---

> > > ### Author Response · Authors · 2026-04-06
> > >
> > > We thank the reviewer for carefully reading our rebuttal.
> > >
> > > Regarding the theoretical justification for filtering errors, it has been discussed in **Rebuttal Q1 & W1**.
> > > For the justification of the design for the InsLen score, each component is designed to either capture contextual/visual information (e.g., $S_{\rm con}, S_{\rm local}$) or provide a corrective effect (e.g., $S_{\rm cafe}, \overline{p}(o \mid {\hat{z_1}, \ldots, \hat{z_m}})$).
> > > These distance-based metrics and probability-based components are intuitive and easy to interpret.
> > > Therefore, we further discuss simplifying the overall scoring function to enhance the robustness of the InsLen score as follows.
> > >
> > > Beyond the simplification under specific task settings discussed in **Rebuttal Q2 & W2**, we note that **the original InsLen separately uses instruction and image embeddings**, followed by a weighted linear fusion with $\omega$, which introduces additional complexity.
> > > To address this, **our InsLen score can be reformulated to directly integrate information from both instruction and image embeddings**, thereby simplifying the overall formulation.
> > >
> > > First, the original linear combination is replaced by a multiplicative operation as $S_{\rm InsLen}=S_{\rm cls}(\mathbf{o})\cdot S_{ccs}(\mathbf{o})$, eliminating the need for the weighting parameter $\omega$.
> > >
> > > Second, the magnitude scaling term $||h_o||$ and $\alpha$, previously used in $S_{\rm con}=\alpha - \frac{||h_o-\overline{z}||}{||\overline{h_o}||}$ in Eq.(6) to scale the score, can be omitted  under the multiplicative formulation.
> > > As a result, the original formulation $S_{\rm con}(o)$ can be simplified to $-||h_o-\overline{z}||$.
> > > At this point, the scoring function becomes
> > >
> > > $$S_{\rm InsLen}=S_{\rm cls}(\mathbf{o})\cdot\overline{p}(o \mid {\hat{z_1}, \ldots, \hat{z_m}})\cdot (-||h_o-\overline{z}||) \tag{E1}$$
> > >
> > > Moreover, the calibration confidence $S_{\rm cafe}(\mathbf{o})$ in $S_{\rm cls}(\mathbf{o})=S_{\rm local}(\mathbf{o})\cdot S_{\rm cafe}(\mathbf{o})$, defined as the Top-1 confidence assigned to the object over instruction embeddings, can replace $\overline{p}(o \mid {\hat{z_1}, \ldots, \hat{z_m}})$ in Eq.(7) for simplification and is shared with the calibrated local score.
> > > Building on this, the overall scoring function is refined as
> > >
> > > $$S_{\rm InsLen-v2} = \overbrace{S_{\text{local}}(\mathbf{o})\cdot S_{{\rm cafe}}}^{S_{\rm cls}(\mathbf{o})~\text{in Eq.(5)}}
> > > \underbrace{(\mathbf{o})\cdot (-||h_o-\overline{z}||)}_{\text{revised CCS score}}.
> > > \tag{E2}$$
> > >
> > > Therefore, the score in Eq.(E2) consists of three components, and the only remaining hyperparameter is $m$, which controls the selection of the top-$m$ most relevant instruction embeddings.
> > > As shown in Table R11, the revised score, termed InsLen-v2, remains highly effective and consistently outperforms the strongest baselines.
> > >
> > > *Table R11: Comparison on MSCOCO benchmark and CLEVR benchmark.*
> > > |Benchmark|Method|AUROC|AUPR|AUROC|AUPR|
> > > |-|-|-:|-:|-:|-:|
> > > |||LLaVA||Qwen3-VL||
> > > ||GLSIM|83.2|93.8|73.3|93.4|
> > > ||SVAR|73.9|91.3|75.3|93.4|
> > > |MSCOCO|InsLen|**86.9**|**96.3**|**81.0**|**96.7**|
> > > ||InsLen-v2|$\underline{84.7}$|$\underline{95.7}$|$\underline{79.6}$|$\underline{95.4}$|
> > > |||LLaVA||Qwen3-VL||
> > > ||GLSIM|50.3|49.8|62.7|97.9|
> > > ||SVAR|50.8|44.2|73.0|$\underline{98.6}$|
> > > |CLEVR|InsLen|**62.3**|**55.3**|**77.7**|**98.9**|
> > > ||InsLen-v2|$\underline{61.3}$|$\underline{54.2}$|$\underline{75.3}$|98.1|
> > >
> > > As suggested, we will include a discussion of the InsLen-v2 variant in the revised manuscript to provide a simpler scoring function for implementation.

---

### Official Review · Reviewer_Uxuf · 2026-03-10

**Soundness:** 3
**Presentation:** 3
**Significance:** 2
**Originality:** 3
**Overall Recommendation:** 4
**Confidence:** 4

**Summary:**

This paper proposes the Instruction Lens Score (InsLen), a plug-and-play object hallucination detector for multimodal large language models (MLLMs). The key insight is that instruction token embeddings implicitly encode visual information while filtering out misleading visual signals. InsLen combines a Calibrated Local Score (which calibrates vision-based confidence using instruction embeddings) with a Context Consistency Score (which measures agreement between generated object tokens and object-related instruction embeddings). The method requires no auxiliary models or additional training and is evaluated across multiple MLLM architectures and hallucination benchmarks.

**Compliance With Llm Reviewing Policy:**

Affirmed.

**Final Justification:**

my concerns are sufficiently addressed.

**Key Questions For Authors:**

1. How sensitive is InsLen to the specific instruction template used? Have you tested with significantly different instruction formats (e.g., single-word prompts vs. detailed multi-sentence instructions)?
2. What is the computational overhead of InsLen relative to the base MLLM inference? How does it scale with instruction length and number of generated objects?
3. Can you provide a failure case analysis? What types of hallucinations does InsLen consistently miss, and why?

**Limitations:**

No. The paper has no Limitations section and no Impact Statement. The conclusion (Section 6) is purely positive with no acknowledgment of the method's scope or failure modes.

**Strengths And Weaknesses:**

## Strengths
- The observation that instruction embeddings filter misleading visual information is genuinely interesting and well-supported by the Logit Lens analysis (Figure 1, Figure 2). This is a previously overlooked signal for hallucination detection.
- The method requires no auxiliary models, external data, or fine-tuning, making it highly practical for deployment across different MLLM architectures.
- Experiments span multiple MLLMs (LLaVA-1.5, LLaVA-OneVision1.5, Qwen3-VL, mPLUG-Owl3) and benchmarks (MSCOCO, etc.), showing consistent improvements on AUROC over prior methods.
- The two-component score (calibrated local + context consistency) is logically motivated — one addresses local visual evidence calibration, the other captures global contextual coherence.
- The paper is well-organized with informative figures illustrating the core insight.

## Weaknesses
- The paper does not cite or discuss Visual Contrastive Decoding (VCD, Leng et al., 2023), a prominent training-free approach to hallucination mitigation that also leverages contrastive signals from visual inputs. While VCD targets hallucination *mitigation* during generation rather than post-hoc *detection*, the two approaches are closely related in motivation and technique.
- On LLaVA-1.5 and InstructBLIP (Table 3), AUROC values with some baselines below random. The paper acknowledges "limited answer accuracy" but does not sufficiently discuss when and why the method fails.
- The method requires choosing a specific decoder layer per model with no discussion of how these were selected or how sensitive results are to this choice. This is a critical practical deployment concern.
- The paper does not adequately discuss when InsLen fails.
- The method relies on instruction token embeddings, but different prompt templates and instruction styles could significantly affect the filtering behavior. The paper lacks ablation over different instruction formats.
- Computing confidence across all instruction embeddings for every object token could be expensive for long instructions or many generated objects. Computational overhead analysis is insufficient.

---

> ### Author Rebuttal · Authors · 2026-03-31
>
> We thank the reviewer for insightful comments and will address the questions below.
> ## Q1 & W5:Sensitivity to instruction templates.
> We would kindly remind that sensitivity analysis on various instruction templates has been performed (Table 6 in the manuscript, with template details listed in Appendix Table 13), where long instructions use multi-sentence prompts. The results show that longer instructions improve performance by providing richer guidance, while shorter instructions cause a slight drop due to reduced contextual information. As suggested, we evaluate performance on single-word instructions. The results in Table R9 show that the performance exhibits a slight degradation compared to the original instruction. Note that InsLen outperforms the baselines under all settings. Overall, our InsLen is robust to instruction templates.
>
> *Table R9: Evaluation using single-word instructions.*
> |Template|Method|AUROC|AUPR|
> |-|-:|-:|-:|
> |"Caption"|GLSIM|82.1|93.5|
> ||InsLen|85.3|93.7|
> |"Detail"|GLSIM|83.0|93.6|
> ||InsLen|85.2|94.7|
> |"Describe"|GLSIM|81.9|93.7|
> ||InsLen|84.1|93.6|
> |Original instructions|GLSIM|83.2|93.8|
> ||InsLen|86.9|96.3|
>
> ## Q2: Computational overhead.
> As shown in Table R10, the computational cost for detection is much smaller than that of base MLLM inference (row2 vs. row6).
> Given $N_{i}$ instruction embeddings and $N_{v}$ image embeddings, projecting these embeddings into the vocabulary space $\mathcal{V}$ incurs a time complexity of $O((N_i+N_v)* |\mathcal{V}| * d)$, where $d$ denotes the embedding dimension. This indicates that the processing cost grows linearly with the instruction length. For each generated token, the time complexity of Top-K embedding selection is $O(N_{i}\log(N_{i}) + N_{v}\log(N_{v}))$. Therefore, for detecting $m$ generated tokens, the time cost is $O(m\cdot (N_{i}\log(N_{i}) + N_{v}\log(N_{v})))$. These results suggest that the cost for detection is negligible compared to generation.
>
> *Table R10: Runtime per sample with default configurations.*
> |Method|LLaVA(ms)|Qwen3-VL(ms)|
> |-|-:|-:|
> |GLSIM|95.2|537.2|
> |InsLen|104.5|564.5|
> |Top-K image embedding selection per object|0.5|0.7|
> |Image embedding projection|38.5|97.6|
> |Instruction embedding projection|4.5|15.5|
> |Answer generation|1870.0|19550.0|
>
> ## Q3 & W4:Failure analyse.
> The negative effect of Q-Former component for our InsLen score is discussed in Appendix A5.
> Moreover, we observe that detection performance is relatively weak for small objects.
> On MSCOCO benchmark, when distinguishing hallucinated tokens from objects whose area is less than 0.1% of the image, the AUC drops to only 80.27%. This degradation is likely due to the limited visual information contained in small objects, which makes them difficult for the object hallucination detector to reliably identify. Consequently, the overall detection confidence tends to be lower in such cases. Please refer to the response to reviewer bqQy W4, where we show that visually similar hallucinations remain challenging for current detectors.
>
> ## W1: Difference from VCD.
> We appreciate the comment on the relation to VCD.
> However, InsLen and VCD differ in both motivation and technique.
> While both leverage visual signals, they differ in motivation: VCD addresses language priors and statistical bias, whereas our method exploits instruction embeddings to extract robust visual information for reliable detection.
> Technically, VCD corrects error result by comparing output differences from original visual inputs and distorted visual inputs. In different views, our InsLen score leverages information from instruction embeddings for OH detection rather than contrastive signals from visual inputs. As suggested, we will cite and discuss studies about contrastive decoding strategies including VCD and hallucination mitigation techniques in the new version.
>
> ## W2: Clarification on Below-Random AUROC for baselines.
> Evaluation on the CLEVR benchmark (Table 3) focuses on relational and attribution hallucinations. The low performance of prior baselines mainly stems from their reliance on patch-level image embeddings as detection signals.
> These embeddings are limited in capturing complex attributes (e.g., material) and spatial relations, leading to suboptimal performance of baselines.
> Our InsLen leverages instruction embeddings which further process visual information through cross-modal interactions. As a result, instruction embeddings encode semantic concepts such as attributes and relationships, enabling InsLen to effectively detect attribution and relational hallucinations.
>
> ## W3: Sensitivity of layer choice.
> We would kindly remind that we have provided sensitivity analysis for layer choice in Appendix (Lines 598–618).
> The results show that InsLen performs better with instruction embeddings from deeper layers.
> In addition, we analyze the impact of integrating advanced techniques (e.g., DeepStack in Qwen3-VL), and observe that such architectural designs also influence performance.

---

> > ### Author Rebuttal · Reviewer_Uxuf · 2026-04-04
> >
> > Thanks for the thorough response. My concerns are sufficiently addressed.

---

> > > ### Author Response · Authors · 2026-04-06
> > >
> > > We are pleased to have resolved your issue and sincerely appreciate your feedback.

---

### Official Review · Reviewer_bqQy · 2026-03-11

**Soundness:** 3
**Presentation:** 3
**Significance:** 3
**Originality:** 3
**Overall Recommendation:** 4
**Confidence:** 4

**Summary:**

This paper proposes Instruction Lens Score (InsLen), a training-free method for detecting object hallucinations in multimodal large language models (MLLMs). The authors analyze instruction token embeddings and observe that they implicitly encode visual semantics and can suppress misleading signals from image embeddings. The authors strive to explore a central context in which instruction embeddings act as a reliable signal for identifying hallucinated objects during multimodal generation.

Based on this observation, the authors outline an important concept: instruction embeddings can be used to calibrate visual evidence and provide contextual cues for hallucination detection. The proposed InsLen score combines two components: (1) Calibrated Local Score, which adjusts vision-based similarity scores using confidence derived from instruction embeddings, and (2) Context Consistency Score, which measures the consistency between generated object token embeddings and aggregated instruction embeddings associated with that object.

The final score integrates these two signals to determine whether a generated object token corresponds to a real object in the image. The method is plug-and-play and does not require additional training or external models. Experiments on multiple benchmarks and several MLLMs show that InsLen consistently outperforms existing hallucination detection approaches.

**Compliance With Llm Reviewing Policy:**

Affirmed.

**Final Justification:**

The authors have adequately addressed the major technical concerns ann I would like to maintain the score.

**Key Questions For Authors:**

Sensitivity to instruction prompts.
Since the proposed method relies on instruction embeddings, how sensitive is InsLen to different prompt formulations or instruction styles? Would the performance degrade when using shorter, paraphrased, or task-specific prompts?

Generalization beyond object hallucination.
The current method mainly targets object-level hallucinations. Can the proposed approach be extended to detect other types of hallucinations, such as OCR?

Computational overhead.
The method requires extracting intermediate embeddings and computing additional similarity scores. What is the inference-time overhead compared to standard generation?

Failure cases.
Could the authors provide more qualitative analysis or failure cases where instruction embeddings fail to detect hallucinations, particularly when hallucinated objects are visually similar to real ones?

**Limitations:**

The paper briefly discusses some limitations, but the discussion could be further strengthened. In particular, the method mainly focuses on object-level hallucination detection, and its applicability to other types of multimodal hallucinations (e.g., attribute errors, relational mistakes, or OCR-related textual hallucinations) is not explored. In addition, the reliance on instruction embeddings may introduce sensitivity to prompt design, which could affect robustness in real-world deployments. A more explicit discussion of these limitations and potential impacts would improve the paper.

**Strengths And Weaknesses:**

### Strengths

Novel insight. The paper identifies an interesting and underexplored signal in MLLMs: instruction embeddings can implicitly encode grounded visual information and help suppress misleading visual cues.

Training-free and plug-and-play. The proposed method does not require additional training, external models, or extra supervision, making it easy to integrate with existing MLLMs.

Strong empirical results. Experiments across multiple datasets  and different MLLM architectures demonstrate consistent improvements over several recent baselines.


### Weaknesses

Dependence on prompt structure. Since the method relies on instruction embeddings, performance may vary depending on the instruction wording or prompt length, which raises concerns about robustness in diverse prompting scenarios.

Focus on object hallucination only. The method is mainly designed for object-level hallucinations, and its effectiveness for other hallucination types (e.g., attributes, relationships, reasoning errors, OCR) is less explored.

Limited qualitative analysis. The paper would benefit from more detailed case studies illustrating when and why instruction embeddings succeed or fail in detecting hallucinations.

---

> ### Author Rebuttal · Authors · 2026-03-31
>
> We thank the reviewer for the constructive feedback and address each point below.
> ## W1 & Q1: Sensitivity to instruction styles, length and formulation.
> We would kindly remind that some sensitivity study on  instruction styles/length has been summarized in Table 6 and discussed in the main text, which shows that long instructions in a "look step by step'' manner improve performance, while shorter instructions cause a slight drop.
> We further evaluate single-word instructions in Table R4 and observe comparable performance with short instructions.
>
> *Table R4: Evaluation using single-word instructions.*
> |Template|Method|AUROC|AUPR|
> |-|-:|-:|-:|
> |"Caption"|GLSIM|82.1|93.5|
> ||InsLen|85.3|93.7|
> |"Detail"|GLSIM|83.0|93.6|
> ||InsLen|85.2|94.7|
> |"Describe"|GLSIM|81.9|93.7|
> ||InsLen|84.1|93.6|
> |Short instructions|GLSIM|82.9|93.7|
> ||InsLen|85.6|94.8|
>
> As suggested, we study effect of paraphrased instructions with the same semantics.
> Results remain stable as shown in Table R5, indicating robustness to instruction variations.
> Additionally, please refer to our response to reviewer cEfs W3, where we show that InsLen is also robust to noisy instructions.
>
> *Table R5: Comparison using paraphrased instructions.*
> |Template|Method|AUROC|AUPR|
> |-|-:|-:|-:|
> |"What is shown in the image?"|GLSIM|83.1|95.5|
> ||InsLen|86.3|96.7|
> |"Explain what you see in the picture."|GLSIM|83.0|95.6|
> ||InsLen|85.2|95.7|
> |"Given a detailed description of the scene."|GLSIM|84.9|95.7|
> ||InsLen|86.1|96.6|
>
> For task-specific instructions, evaluations on POPE (binary questions as “Is there a {object} in the image?”) and CLEVR (compositional reasoning prompts tailored to each image) is shown in Table 2 and Table 3.
> Despite these changes in instruction style and task formulation, InsLen outperforms baselines across both benchmarks.
> ## Q2 & W2:Generalization beyond object hallucination.
> We would kindly remind that the evaluation on CLEVR benchmark (Table 3 in the original manuscript) actually considers attribute and relational errors.
> The results show that InsLen remains effective in multiple scenarios beyond object hallucination.
>
> For OCR tasks, we evaluate InsLen with 1,000 examples sampled from OCRBench [c1].
> We perform character-level hallucination detection as OCR tasks require prediction for each character.
> The results in Table R6 show that InsLen achieves the best performance.
>
> *Table R6: Evaluation on OCRBench.*
> |Method|AUROC|AUPR|AUROC|AUPR|
> |-|-:|-:|-:|-:|
> ||LLaVA ||Qwen3-VL||
> |GLSIM|72.6|75.0|81.6|92.3|
> |SVAR|73.6|79.2|80.3|91.6|
> |InsLen|79.5|79.0|84.2|93.2|
>
> Thanks for the suggestion about reasoning errors.
> Reasoning errors lack token-level ground truth for evaluation and may arise despite correct visual grounding, thus beyond our scope.
> We will explore this direction in future work.
> ## W3.Computational overhead.
> The computational overhead of InsLen is negligible compared to generation.
> As shown in Table R7, while InsLen introduces extra operations, these steps are lightweight compared to the generation.
> Moreover, our InsLen uses intermediate embeddings produced during generation, which avoids additional computation cost.
>
> *Table R7: Computational cost with default configurations.*
> |Component|LLaVA (ms)|Qwen3-VL(ms)|
> |-|-:|-:|
> |InsLen|104.5|564.5|
> |Top-K image embedding selection per object|0.5|0.7|
> |Projecting image embeddings|38.5|97.6|
> |Projecting instruction embeddings|4.5|15.5|
> |Answer generation|1870.0|19550.0|
>
> ## W4.Failure analyse.
> We observe that InsLen performs worse on small objects (area $\leq$ 0.1%), with an AUC of 80.27% compared to 86.93% on the full MSCOCO dataset.
> As small objects contain limited visual information, it is challenging to capture such small object information [c2] for current detectors.
>
> To study visually similar hallucinations, we evaluate on Flower102 [c3], using the template "What is the image about? {$n$ options}" with one of the options is the ground truth.
> Results are shown in Table R8 with $n=10$, where our InsLen achieves the best performance.
> However, lower performance on Flower102 compared to MCOCO reveals challenges in fine-grained scenarios.
> This is because MLLMs struggle to capture fine-grained details [c4], providing less discriminative signals for hallucination detection in such cases.
>
> *Table R8: Performance on the Flower102.*
> |Method|AUROC|AUPR|AUROC|AUPR|
> |-|:-|:-|:-|:-|
> ||LLaVA ||Qwen3-VL||
> |GLSIM|72.0|79.2|76.3|83.9|
> |SVAR|71.8|78.9|75.2|82.3|
> |InsLen|74.4|79.7|78.0|84.6|
>
> [c1] OCRBench: On the Hidden Mystery of OCR in Large Multimodal Models.  Science China Information Sciences, 220102, 2024.
>
> [c2] Exploring Perceptual Limitations of Multimodal LLMs on Small Visual Objects. TMLR, 2026.
>
> [c3] Automated flower classification over a large number of classes. In: Indian conference on computer vision, graphics & image processing, 2008.
>
> [c4] Hallucination at a Glance: Controlled Visual Edits and Fine-Grained Multimodal Learning. arXiv:2506.07227, 2025.

---

> > ### Author Rebuttal · Reviewer_bqQy · 2026-04-01
> >
> > Thank you for the detailed and well-organized rebuttal. The additional experiments and clarifications, especially on instruction sensitivity, generalization beyond object hallucination, and computational overhead, have addressed most of my concerns.
> >
> > I find the analysis more complete now, and the empirical evidence is convincing overall. Based on the rebuttal, I am comfortable maintaining my positive score.

---

> > > ### Author Response · Authors · 2026-04-02
> > >
> > > Thank you for reviewing our rebuttal and for your positive feedback. We also appreciate your helpful suggestion.

---

### Official Review · Reviewer_cEfs · 2026-03-12

**Soundness:** 3
**Presentation:** 3
**Significance:** 3
**Originality:** 3
**Overall Recommendation:** 4
**Confidence:** 3

**Summary:**

This paper proposes Instruction Lens Score (InsLen), a training-free, plug-and-play detector for object hallucinations (OH) in multimodal LLMs. The key insight is that instruction token embeddings, when examined via the Logit Lens, implicitly encode image-grounded visual semantics and tend to downweight misleading signals from image embeddings. InsLen combines (i) a Calibrated Local Score that modulates vision-based local evidence with an instruction-derived calibration confidence, and (ii) a Context Consistency Score that measures the consistency between generated object-token embeddings and an aggregate of instruction embeddings most supportive of that object. Across five MLLMs and multiple benchmarks (MSCOCO, Objects365, POPE, CLEVR), InsLen outperforms state-of-the-art detection methods.

**Compliance With Llm Reviewing Policy:**

Affirmed.

**Final Justification:**

The authors have adequately addressed the major technical concerns ann I would like to maintain the score of 4 (Weak Accept).

**Key Questions For Authors:**

none

**Limitations:**

yes

**Strengths And Weaknesses:**

### Strengths
1. Leveraging instruction token embeddings for OH detection is a fresh angle compared to prior methods that focus on visual tokens, attention, or global-local image-text similarities.
2. Results span diverse MLLMs (LLaVA-1.5, InstructBLIP, mPLUG-Owl3, LLaVA-OneVision, Qwen3-VL) and multiple settings (caption-style OH, yes/no POPE, attribute/relational CLEVR), demonstrating robustness and generality.
3. Ablations disentangle contributions of the calibration factor and the context-consistency weighting, and sensitivity analyses examine the trade-off parameter ω and number m of instruction embeddings.
### Weakness
1. The reliance on unembedding projections (Logit Lens) to interpret instruction embeddings assumes the language head is a faithful probe of cross-modal content; this interpretability proxy can be noisy or layer-dependent.
2. The context consistency uses an L2-based score with a hand-tuned α, raising concerns about scale sensitivity and cross-model comparability; cosine or learned Mahalanobis variants are not explored.
3. Potential vulnerability to instruction manipulation: prompts that implicitly or explicitly name objects could inflate S_cafe/S_ccs and weaken hallucination detection.
4. Limited analysis of layer choice and temperature τ; results may depend on the penultimate layer and τ=10, but these sensitivities are not reported.
5. Runtime/compute overhead versus baselines (especially for projecting all instruction tokens and computing Top-K image embeddings per object) is not quantified.
6. POPE adaptation details are under-specified (e.g., how token-level scoring maps to yes/no answers and whether negative questions bias S_cafe).

---

> ### Author Rebuttal · Authors · 2026-03-31
>
> We thank the reviewer for the insightful comment and address the concern below.
> ## W1. Reliability and layer robustness of Logit Lens as a probe for instruction embeddings.
>
> Since answer embeddings integrate visual information through attention mechanism, the language head is able to decode embeddings that fuse textual and visual features.
> Instruction embeddings are initially derived from instruction text tokens and progressively fuse with visual information, and are processed in the same manner as answer embeddings.
> Thus, the language head can also probe instruction embeddings.
> Please refer to Appendix Figure 14-16, where we provide the probing visualization cases.
> The ablation study on the selection of instruction feature layers is conducted in Appendix A2, which shows that instruction embeddings from deeper layers can be effectively decoded by the language head.
>
> ## W2.Robustness of the L2-based context consistency score and its hyperparameter sensitivity.
> The ablation study on the hyperparameter $\alpha$ is provided in the Appendix L619-L629 using LLaVA and Qwen3-VL.
> The results show that the InsLen score is not sensitive to the choice of $\alpha$'s value, alleviating concerns about hand-tuning.
> Additionally, as defined in Eq.(6), the consistency score is computed as $||\overline{z}-h_o||/||h_o||$, where $||h_o||$ normalizes the scale of the score.
> This formulation helps reduce scale sensitivity, thereby improving cross-model comparability.
> For more variants, we would kindly remind that we have included experiments with a cosine similarity variant in Appendix Table 12.
> As suggested, experiments with a Mahalanobis-based score were performed, and the results are shown in Table R1.
> The Mahalanobis-based score achieves performance comparable to the L2-based score.
>
> *Table R1: Performance comparison of our InsLen score under different distance metrics.*
> |Model|Metric|AUROC|AUPR|
> |---|-:|-:|-:|
> |LLaVA|L2|86.9|96.3|
> ||Cos|84.3|95.7|
> ||Mahalanobis|85.7|96.0|
> |Qwen3-VL|L2|81.0|96.7|
> ||Cos|79.3|96.2|
> ||Mahalanobis|81.3|96.5|
>
> ## W3. Impact of object-revealing instructions on InsLen.
> As suggested, we examine the robustness of our InsLen to object-revealing instructions by:
> (1) Inserting object names that do not belong to the image in the text;
> (2) Inserting semantically inconsistent concepts that are irrelevant to the image (e.g., for an image of a “cat,” inserting “fruits”).
> The results are shown in Table R2, which demonstrates that InsLen is robust to instruction manipulation, regardless of whether the object name is inserted randomly, at the beginning, or at the end of the instructions.
>
> *Table R2: Performance under different insertion positions. Condition (1) is shown on the left of '/', and condition (2) on the right.*
> |Model|Position|InsLen|GLSIM|SAVR|
> |---|-|-:|-:|-:|
> |LLaVA|End|85.4/86.0|82.0/82.4|74.2/73.7|
> ||Begin|86.5/86.9|83.3/82.9|73.5/73.9|
> ||Random|85.7/86.3|82.8/83.1|74.0/74.5|
> ||Original instruction|86.9|83.2|74.0|
> |Qwen3-VL|End|81.2/80.7|71.8/73.3|75.0/74.3|
> ||Begin|80.8/80.1|72.1/74.6|74.7/74.2|
> ||Random|80.3/81.8|72.1/74.6|75.7/75.1|
> ||Original instruction |81.0|73.3|75.4|
>
> ## W4. sensitivity to layer selection and temperature $\tau$.
> We would kindly remind that we have provided ablation studies on the temperature coefficient and layer choice in Appendix L598-618.
> The results show that: (1) the InsLen score achieves better separation between hallucinated and truth tokens when temperature values increase from 1 to 30, and the performance becomes  stable as the temperature further increases; (2) deeper layers generally lead to better performance, although certain architectural designs (e.g., Deepstack in Qwen3-VL) may disrupt semantic continuity and introduce performance fluctuations.
>
> ## W5. Computational overhead of InsLen compared to baselines.
> Our InsLen incurs slightly higher computational overhead compared to several baselines, as shown in Table R3.
> However, the cost is minimal when compared to answer generation, and thus still meets the requirements for inference-time detection.
> Please refer to the response to reviewer Uxuf Q3, where we quantify the cost for projecting embeddings and top-k selection.
>
> *Table R3: Runtime analysis per sample.*
> |Method|LLaVA(ms)|Qwen3-VL(ms)|
> |---|-:|-:|
> |SVAR|12.9|15.7|
> |GLSIM|95.2|537.2|
> |EASY|3896.7|40293.3|
> |InsLen|104.5|564.5|
> |Answer generation|1870.0|19550.0|
>
> ## W6. Clarifying the POPE adaptation and potential bias.
> In the POPE benchmark, we compute the confidence score on the "Yes/No" tokens following prior work GLSIM.
> Specifically, for example, the calibration confidence is computed as $S_{cafe}(\text{"Yes''}) = \max_j \ \mathrm{softmax}(W_u \cdot z_j)[\text{"Yes''}]$.
> Therefore, we can directly measure the confidence of the "Yes/No" tokens.
> Negative questions don't bias $S_{cafe}$.
> Given the input instruction, the score is computed over confidence on “Yes/No” tokens; it remains largely unaffected by the presence of negative objects.

---

> > ### Author Rebuttal · Reviewer_cEfs · 2026-04-05
> >
> > Thanks for the authors thorough rebuttal. The responses to W2 (distance metric comparison), W4 (layer and τ sensitivity), and W5 (runtime analysis) are convincing and fully address our concerns. The additional experiments on instruction manipulation (W3) also provide useful evidence of robustness.
> >
> > Regarding W1, the visualization and ablation provide reasonable support, though a more systematic analysis of when Logit Lens fails as a probe would further strengthen the paper. That beining said, the authors have adequately addressed the major technical concerns. So I would like to maintain the score of 4 (Weak Accept).

---

> > > ### Author Response · Authors · 2026-04-06
> > >
> > > We’re glad we could resolve your concerns, and we truly appreciate your feedback.
> > >
> > > As suggested, we will discuss the potential impact of the Logit Lens limitations on our InsLen score in the revised manuscript, including the representation drift issue [c9], and explore potential improvements based on prior work [c9, c10].
> > >
> > > [c9] Eliciting Latent Predictions from Transformers with the Tuned Lens. arXiv:2303.08112, 2023.
> > >
> > > [c10] Looking Beyond the Top-1: Transformers Determine Top Tokens in Order. In: ICML, 2025.

---

### Decision · Program_Chairs · 2026-04-30

**Decision:**

Accept (regular)

**Comment:**

The main tension comes from reviewer V8kB for requesting of the theoretical analysis. Frankly speaking, I think the reviewer should be more specific about what kind of theoretical analysis is required. Based on my understanding the reviewer might require a logic reasoning why a generic instruction such as “Please describe the image in detail.” may reflect language priors more strongly than actual visual evidence. The reviewer might need a more concrete reasoning argument (not empirical evidences) However the author might understand this as finding a theoretical background for the proposed approach.

Frankly speaking, I think this reviewer comment is not very well constructed and the answer might not fully address the reviewer's comment. However I don't think the author should be punished based on this. And all the other three reviewers vote for weak accept. I agree with them and vote for weak accept.